# ConserWeightive Behavioral Cloning for Reliable Offline Reinforcement Learning

## Abstract

The goal of offline reinforcement learning (RL) is to learn near-optimal policies from static logged datasets, thus sidestepping expensive online interactions. Behavioral cloning (BC) provides a straightforward solution to offline RL by mimicking offline trajectories via supervised learning. Recent advances (Chen et al., 2021; Janner et al., 2021; Emmons et al., 2021) have shown that by conditioning on desired future returns, BC can perform competitively to their value-based counterparts, while enjoying much more simplicity and training stability. However, the distribution of returns in the offline dataset can be arbitrarily skewed and suboptimal, which poses a unique challenge for conditioning BC on expert returns at test-time. We propose ConserWeightive Behavioral Cloning (CWBC), a simple and effective method for improving the performance of conditional BC for offline RL with two key components: trajectory weighting and conservative regularization. Trajectory weighting addresses the bias-variance tradeoff in conditional BC and provides a principled mechanism to learn from both low return trajectories (typically plentiful) and high return trajectories (typically few). Further, we analyze the notion of conservatism in existing BC methods, and propose a novel conservative regularizer that explicitly encourages the policy to stay close to the data distribution. The regularizer helps achieve more reliable performance, and removes the need for ad-hoc tuning of the conditioning value during evaluation. We instantiate CWBC in the context of Reinforcement Learning via Supervised Learning (RvS) (Emmons et al., 2021) and Decision Transformer (DT) (Chen et al., 2021), and empirically show that it significantly boosts the performance and stability of prior methods on various offline RL benchmarks.

## 1 Introduction

In many real-world applications such as education, healthcare and autonomous driving, collecting data via online interactions can be expensive or even dangerous. However, we often have access to historical logged datasets in these domains that have been collected previously by some unknown policies. The goal of offline reinforcement learning (RL) is to directly learn effective agent policies from such datasets, without additional online interactions (Lange et al., 2012; Levine et al., 2020). Many online RL algorithms have been adapted to work in the offline setting, including value-based methods (Fujimoto et al., 2019; Ghasemipour et al., 2021; Wu et al., 2019; Jaques et al., 2019; Kumar et al., 2020; Fujimoto & Gu, 2021; Kostrikov et al., 2021a) as well as model-based methods (Yu et al., 2020; Kidambi et al., 2020). The key challenge in all these methods is to generalize the value or dynamics to state-action pairs outside the offline dataset.

An alternative way to approach offline RL is via approaches derived from behavioral cloning (BC) (Bain & Sammut, 1995). BC is a supervised learning technique that was initially developed for imitation learning, where the goal is to learn a policy that mimics the expert demonstrations. Recently, a number of works propose to formulate offline RL as supervised learning problems (Chen et al., 2021; Janner et al., 2021; Emmons et al., 2021). Since offline RL datasets usually do not have expert demonstrations, these works condition BC on extra context information to specify target outcomes such as returns and goals. Compared with the value-based approaches, the empirical evidence has shown that these conditional BC approaches perform competitively, and they additionally enjoy the enhanced simplicity and training stability of supervised learning.

As commonly observed for supervised learning approaches, the performance of conditional BC is often limited by the suboptimility of the offline dataset, which particularly can be probed through the distribution of returns in the dataset. There are two related challenges in this regard for offline RL.

First, there is a unique bias-variance tradeoff in learning that arises due to the mismatch between the training and test distribution of returns. Typically, offline datasets in the real world mostly contain trajectories with low returns, whereas at test time, we are interested in conditioning on high returns. Simply filtering the offline dataset to contain high return trajectories is not always viable, as the number of such high-return trajectories can be very low leading to high variance during learning.

Second, the maximum return in the offline trajectories is often far below the desired expert returns. This implies that at test time, we need to condition our agent on out-of-distribution (ood) expert returns. Interestingly, we find that existing BC methods have significantly different behaviors when conditioning on ood returns. While DT (Chen et al., 2021) enjoys a stable performance, RvS (Emmons et al., 2021) is highly sensitive to such ood conditioning and exhibits vast drops in peak performance for such ood inputs. Therefore, the current practice for setting the conditioning return at test-time in RvS is based on careful tuning with online rollouts, which is often tedious, impractical, and inconsistent with the promise of offline RL to minimize online interactions.

We propose ConserWeightive Behavior Cloning (CWBC), a new BC-based approach for offline RL that mitigates the aforementioned challenges. CWBC consists of 2 key components: trajectory weighting and conservative regularization. With trajectory weighting, we strive to balance the bias-variance trade-off in learning by proposing a scheme for downweighting the low-return trajectories, but at the same time, we do not filter them for data efficiency. Moreover, we introduce a notion of *conservatism* for ood sensitve BC methods such as RvS, which encourages the policy to stay close to the data distribution when conditioning on large returns. We take trajectories with high returns from the dataset and add positive noise to their returns, which generates trajectories with large ood returns. We predict actions conditioning on the perturbed returns and project them to the original actions by penalizing the $\ell_2$ distance. By imposing such a regularizer, we can condition the policy on large, unseen target returns at test-time, sidestepping tedious manual tuning and online interactions.

Our proposed algorithm is simple and easy to implement. Empirically, we instantiate our framework in the context of RvS (Emmons et al., 2021) and DT (Chen et al., 2021), two state-of-the-art BC methods for offline RL. CWBC significantly improves the performance of RvS and DT in D4RL (Fu et al., 2020) locomotion tasks by $18\%$ and $8\%$, respectively, without any hand picking of the value of the conditioning returns at test-time.

## 2  RELATED WORK

**Offline Temporal Difference Learning**   Most of the existing off-policy RL methods are often based on temporal difference (TD) updates. A key challenge of directly applying them in the offline setting is the *extrapolation error*: the value function is poorly estimated at unseen state-action pairs. To remedy this issue, various forms of *conservatism* have been introduced to off-policy RL methods that exploits temporal difference updates, with the purpose of encouraging the learned policy to stay close to the behavior policy that generates the data. For instance, Fujimoto et al. (2019); Ghasemipour et al. (2021) use certain policy parameterizations specifically tailored for offline RL. Wu et al. (2019); Jaques et al. (2019); Kumar et al. (2019) penalize the divergence-based distances between the learned policy and the behavior policy. Fujimoto & Gu (2021) propose an extra behavior cloning term to regularize the policy. This regularizer is simply the $\ell_2$ distance between predicted actions and the truth, yet surprisingly effective for porting off-policy TD methods to the offline setting. Instead of regularizing the policy, several other works have sought to incorporate divergence regularizations into the value function estimation, e.g., (Nachum et al., 2019; Kumar et al., 2020; Kostrikov et al., 2021a). Another recent work by Kostrikov et al. (2021b) predicts the $Q$ function via expectile regression, where the estimation of the maximum $Q$-value is constrained to be in the dataset.

**Behavior Cloning Approaches for Offline RL**   Recently, there is a surge of interest in converting offline RL into supervised learning paradigms (Chen et al., 2021; Janner et al., 2021; Emmons et al., 2021). In essence, these approaches conduct behavior cloning (Bain & Sammut, 1995) by additionally conditioning on extra information such as goals or rewards. Among these works, Chen et al. (2021) and Janner et al. (2021) have formulated offline RL as sequence modeling problems

and train transformer architectures (Vaswani et al., 2017) in a similar fashion to language and vision (Radford et al., 2018; Chen et al., 2020; Brown et al., 2020; Lu et al., 2022; Yan et al., 2021). Extensions have also been proposed in the context of sequential decision making for offline black-box optimization (Nguyen & Grover, 2022; Krishnamoorthy et al., 2022). A recent work by Emmons et al. (2021) further shows that conditional BC can achieve competitive performance even with a simple but carefully designed MLP network. Earlier, similar ideas have also been proposed for online RL, where the policy is trained via supervised learning techniques to fit the data stored in the replay buffer (Schmidhuber, 2019; Srivastava et al., 2019; Ghosh et al., 2019).

**Data Exploration for Offline RL**   Recent research efforts have also been made towards understanding properties and limitations of datasets used for offline RL (Yarats et al., 2022; Lambert et al., 2022; Guo et al., 2021), particularly focusing on exploration techniques during data collection. Both Yarats et al. (2022) and Lambert et al. (2022) collect datasets using task-agnostic exploration strategies (Laskin et al., 2021), relabel the rewards and train offline RL algorithms on them. Yarats et al. (2022) benchmark multiple offline RL algorithms on different tasks including transferring, whereas Lambert et al. (2022) focus on improving the exploration method.

## 3   PRELIMINARIES

We model our environment as a Markov decision process (MDP) (Bellman, 1957), which can be described by a tuple $\mathcal{M} = \langle \mathcal{S}, \mathcal{A}, p, P, R, \gamma \rangle$, where $\mathcal{S}$ is the state space, $\mathcal{A}$ is the action space, $p(s_1)$ is the distribution of the initial state, $P(s_{t+1}|s_t, a_t)$ is the transition probability distribution, $R(s_t, a_t)$ is the deterministic reward function, and $\gamma$ is the discount factor. At each timestep $t$, the agent observes a state $s_t \in S$ and takes an action $a_t \in \mathcal{A}$. This moves the agent to the next state $s_{t+1} \sim P(\cdot|s_t, a_t)$ and provides the agent with a reward $r_t = R(s_t, a_t)$.

**Offline RL.**   We are interested in learning a (near-)optimal policy from a static offline dataset of trajectories collected by unknown policies, denoted as $\mathcal{T}_{\text{offline}}$. We assume that these trajectories are *i.i.d* samples drawn from some unknown static distribution $\mathcal{T}$. We use $\tau$ to denote a trajectory and use $|\tau|$ to denote its length. Following Chen et al. (2021), the return-to-go (RTG) for a trajectory $\tau$ at timestep $t$ is defined as the sum of rewards starting from $t$ until the end of the trajectory: $g_t = \sum_{t'=t}^{|\tau|} r_{t'}$. This means the initial RTG $g_1$ is equal to the total return of the trajectory $r_\tau = \sum_{t=1}^{|\tau|} r_t$.

**Decision Transformer (DT).**   DT (Chen et al., 2021) solves offline RL via sequence modeling. Specifically, DT employs a transformer architecture that generates actions given a sequence of historical states and RTGs. To do that, DT first transforms each trajectory in the dataset into a sequence of returns-to-go, states, and actions:

$$\tau = \left( g_1, s_1, a_1, g_2, s_2, a_2, \ldots, g_{|\tau|}, s_{|\tau|}, a_{|\tau|} \right). \tag{1}$$

DT trains a policy that generates action $a_t$ at each timestep $t$ conditioned on the history of RTGs $g_{t-K:t}$, states $s_{t-K:t}$, and actions $a_{t-K:t-1}$, wherein $K$ is the context length of the transformer. The learning objective a simple mean square error between the predicted actions and the ground truths:

$$\min_\theta \mathcal{L}_{\text{DT}}(\theta) = \mathbb{E}_{\tau \sim \mathcal{T}} \left[ \frac{1}{|\tau|} \sum_{t=1}^{|\tau|} \left( a_t - \pi_\theta(g_{t-K:t}, s_{t-K:t}, a_{t-K:t-1}) \right)^2 \right]. \tag{2}$$

During evaluation, DT starts with an initial state $s_1$ and a target RTG $g_1$. At each step $t$, the agent generates an action $a_t$, receives a reward $r_t$ and observes the next state $s_{t+1}$. DT updates its RTG $g_{t+1} = g_t - r_t$ and generates next action $a_{t+1}$. This process is repeated until the end of the episode.

**Reinforcement Learning via Supervised Learning (RvS).**   Emmons et al. (2021) conduct a thorough empirical study of conditional BC methods under the umbrella of Reinforcement Learning via Supervised Learning (RvS), and show that even simple models such as multi-layer perceptrons (MLP) can perform well. With carefully chosen architecture and hyperparameters, they exhibit performance that matches or exceeds the performance of transformer-based models. There are two main differences between RvS and DT. First, RvS conditions on the average reward $\omega_t$ into the future instead of the sum of future rewards:

$$\omega_t = \frac{1}{H-t+1} \sum_{t'=t}^{|\tau|} r_{t'} = \frac{g_t}{H-t+1}, \tag{3}$$

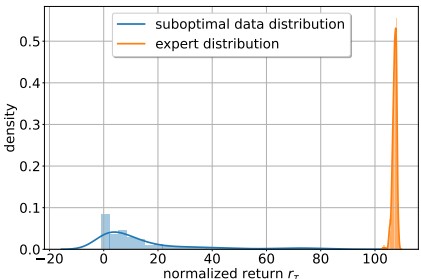 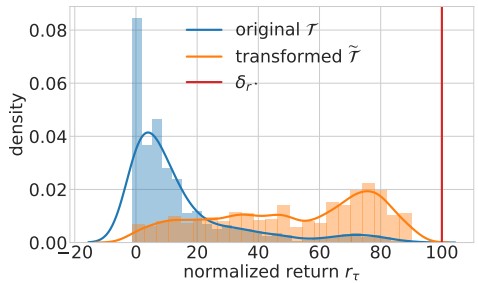

(a) Offline data distribution vs the expert distribution.

(b) The original return distribution $\mathcal{T}$ and the transformed distribution $\widetilde{\mathcal{T}}$.

Figure 1: The suboptimality of offline datasets (left) and the effect of trajectory weighting on the return distribution (right). We illustrate on `walker2d-med-replay`. For weighting, we use $B = 20$, $\lambda = 0.01$, $\kappa = \widehat{r}^{\star} - \widehat{r}_{90}$, where $\widehat{r}_{90}$ is the 90-th percentile of the returns in the offline dataset.

where $H$ is the maximum episode length. Intuitively, $\omega_t$ is RTG normalized by the number of remaining steps. Second, RvS employs a simple MLP architecture, which generates action $a_t$ at step $t$ based on only the current state $s_t$ and expected outcome $\omega_t$. RvS minimizes a mean square error:

$$\min_{\theta} \mathcal{L}_{\text{RvS}}(\theta) = \mathbb{E}_{\tau \sim \mathcal{T}} \left[ \frac{1}{|\tau|} \sum_{t=1}^{|\tau|} \left( a_t - \pi_{\theta}(s_t, \omega_t) \right)^2 \right]. \tag{4}$$

At evaluation time, RvS performs a repeating process similarly to DT, except that the expected outcome is now updated as $\omega_{t+1} = (g_t - r_t)/(H - t)$.

## 4 CONSERVATIVE BEHAVIORAL CLONING WITH TRAJECTORY WEIGHTING

A key challenge that behavioral cloning faces in an offline setting is the suboptimality of the dataset, which we can characterize via the distribution of trajectory returns. An ideal offline dataset consists of sufficiently many high-quality trajectories, which have returns matching those of a dataset of expert demonstrations. For such an idealized scenario, offline RL reduces to a vanilla imitation learning problem. In practice, however, we observe that the return distribution for a typical dataset of offline trajectories is spread over a wide range of returns and is highly non-uniform. Figure 1a illustrates the return distribution of the `walker2d-med-replay` dataset (Fu et al., 2020), which is significantly different from the expert distribution. Therefore, from a return perspective, the trajectories in the offline dataset can be of varying importance for learning, which leads to a bias-variance trade-off. Further, for return-conditioned methods including conditional BC, it is unclear how the policy will behave when conditioned on o.o.d. returns at test-time. We study mitigation techniques for both these challenges in the following sections.

### 4.1 CONTROLLING BIAS-VARIANCE TRADEOFF VIA TRAJECTORY WEIGHTING

To formalize our discussion, recall that $r_\tau$ denotes the return of a trajectory $\tau$ and let $r^\star = \sup_\tau r_\tau$ be the maximum expert return, which is assumed to be known in prior works on conditional BC (Chen et al., 2021; Emmons et al., 2021). We know that the optimal offline data distribution, denoted by $\mathcal{T}^\star$, is simply the distribution of demonstrations rolled out from the optimal policy. Typically, the offline trajectory distribution $\mathcal{T}$ will be biased w.r.t. $\mathcal{T}^\star$. During learning, this leads to a bias-variance tradeoff, wherein ideally we want to learn our BC agent to condition on the expert returns, but is forced to minimize the empirical risk on a biased data distribution.

The core idea of our approach is to transform $\mathcal{T}$ into a new distribution $\widetilde{\mathcal{T}}$ that better estimates $\mathcal{T}^\star$. More concretely, $\widetilde{\mathcal{T}}$ should concentrate on high-return trajectories, which mitigates the bias. One naive strategy is to simply filter out a small fraction of high return trajectories from the offline dataset. However, since we expect the original dataset to contain very few high return trajectories, filtering trajectories will increase the variance for downstream BC. To balance the bias-variance trade-off, we propose to weight the trajectories based on their returns. Let $f_\mathcal{T} : \mathbb{R} \mapsto \mathbb{R}_+$ be the density function of $r_\tau$ where $\tau \sim \mathcal{T}$. We consider the transformed distribution $\widetilde{\mathcal{T}}$ whose density function $p_{\widetilde{\mathcal{T}}}$ is

---
**Algorithm 1:** Weighted Trajectory Sampling

---
1 **Input:** offline dataset $\mathcal{T}_{\text{offline}}$, number of bins $B$, smoothing parameters $\lambda, \kappa$
2 Compute the returns: $r_\tau \leftarrow \sum_{t=1}^{|\tau|} r_t, \ \forall \tau \in \mathcal{T}_{\text{offline}}$.
3 Group the trajectories into $B$ equal-sized bins according to $r_\tau$.
4 Sample a bin $b \in [B]$ with probability $\mathbb{P}_{\text{bin}}(b)$ defined in Equation (6).
5 Sample a trajectory $\tau$ in bin $b$ uniformly at random.
6 **Output:** $\tau$

---

Table 1: The normalized return on D4RL locomotion tasks of RvS and DT with trajectory weighting. We use +W as shorthand for weighting. We use #wins to denote the number of datasets where the variant outperforms the original model. The results are averaged over 10 seeds.

| | DT | DT+W | RvS | RvS+W |
|---|---|---|---|---|
| walker2d-medium | $71.5 \pm 3.9$ | $70.4 \pm 4.5$ | $73.3 \pm 5.7$ | $54.5 \pm 7.7$ |
| walker2d-med-replay | $53.4 \pm 12.2$ | $60.5 \pm 8.9$ | $54.0 \pm 12.1$ | $61.2 \pm 14.7$ |
| walker2d-med-expert | $99.8 \pm 21.3$ | $108.2 \pm 0.8$ | $102.2 \pm 104.1$ | $104.1 \pm 0.5$ |
| hopper-medium | $59.9 \pm 4.9$ | $63.9 \pm 4.4$ | $56.6 \pm 5.5$ | $62.5 \pm 7.1$ |
| hopper-med-replay | $56.4 \pm 20.1$ | $76.9 \pm 5.9$ | $87.7 \pm 9.7$ | $92.4 \pm 6.1$ |
| hopper-med-expert | $95.4 \pm 11.3$ | $103.4 \pm 9.0$ | $108.8 \pm 0.9$ | $108.4 \pm 1.8$ |
| halfcheetah-medium | $42.5 \pm 0.6$ | $41.6 \pm 1.7$ | $16.2 \pm 4.5$ | $4.0 \pm 5.4$ |
| halfcheetah-med-replay | $34.5 \pm 4.2$ | $36.9 \pm 2.2$ | $-0.4 \pm 2.7$ | $-0.8 \pm 2.2$ |
| halfcheetah-med-expert | $87.2 \pm 2.7$ | $85.6 \pm 2.0$ | $83.4 \pm 2.1$ | $69.1 \pm 3.7$ |
| # wins | / | 6 | / | 4 |
| average | 66.7 | 71.9 | 64.6 | 61.7 |

$$p_{\widetilde{\mathcal{T}}}(\tau) \propto \overbrace{\frac{f_\mathcal{T}(r_\tau)}{f_\mathcal{T}(r_\tau)+\lambda} \cdot \exp\left(-\frac{|r_\tau - r^\star|}{\kappa}\right)}^{\text{trajectory weight}}, \tag{5}$$

where $\lambda, \kappa \in \mathbb{R}_+$ are two hyperparameters. A larger value of $\kappa$ leads to a more uniform $\widetilde{\mathcal{T}}$, whereas a smaller value upweights the high-return trajectories. In contrast, a smaller value of $\lambda$ gives more weights to high-return trajectories, while a larger value makes $\widetilde{\mathcal{T}}$ closer to $\mathcal{T}$. Our trajectory weighting is motivated by a similar scheme proposed for model-based optimization (Kumar & Levine, 2020), where the authors use it to balance the bias and variance for gradient approximation for surrogates to black-box functions, and theoretically establish the optimality of the proposed distribution.

### 4.1.1 IMPLEMENTATION DETAILS

In practice, the dataset $\mathcal{T}_{\text{offline}}$ only contains a finite number of samples and the density function $p_{\widetilde{\mathcal{T}}}$ in equation (5) cannot be computed exactly. Following Kumar & Levine (2020), we sample from a discretized approximation of $\widetilde{\mathcal{T}}$. We first group the trajectories in $\mathcal{T}_{\text{offline}}$ into $B$ equal-sized bins according to the return $r_\tau$. To sample a trajectory, we first sample a bin index $b \in \{1, \ldots, B\}$ and then uniformly sample a trajectory inside bin $b$. We use $|b|$ to denote the size of bin $b$. Let $\bar{r}_\tau^b = 1/|b| \sum_{\tau \in b} r_\tau$ the average return of the trajectories in bin $b$, $\hat{r}^\star$ be the highest return in the dataset $\mathcal{T}_{\text{offline}}$, and define $f_{\mathcal{T}_{\text{offline}}}(b) = |b|/|\mathcal{T}_{\text{offline}}|$. As a discretized version of equation (5), the bins are weighted by their average returns with probability

$$\mathbb{P}_{\text{bin}}(b) \propto \frac{f_{\mathcal{T}_{\text{offline}}}(b)}{f_{\mathcal{T}_{\text{offline}}}(b)+\lambda} \cdot \exp\left(-\frac{|\bar{r}_\tau^b - \hat{r}^\star|}{\kappa}\right). \tag{6}$$

Algorithm 1 summarizes the data sampling procedure when trajectory weighting is used. Figure 1b illustrates the impact of trajectory weighting on the return distribution of the med-replay dataset for the walker2d environment. We plot the histograms before and after transformation, where the density curves are estimated by kernel density estimators.

### 4.1.2 EMPIRICAL RESULTS

**Dataset** We evaluate the effectiveness of trajectory weighting on three locomotion tasks with dense rewards from the D4RL benchmark (Fu et al., 2020): hopper, walker2d and halfcheetah.

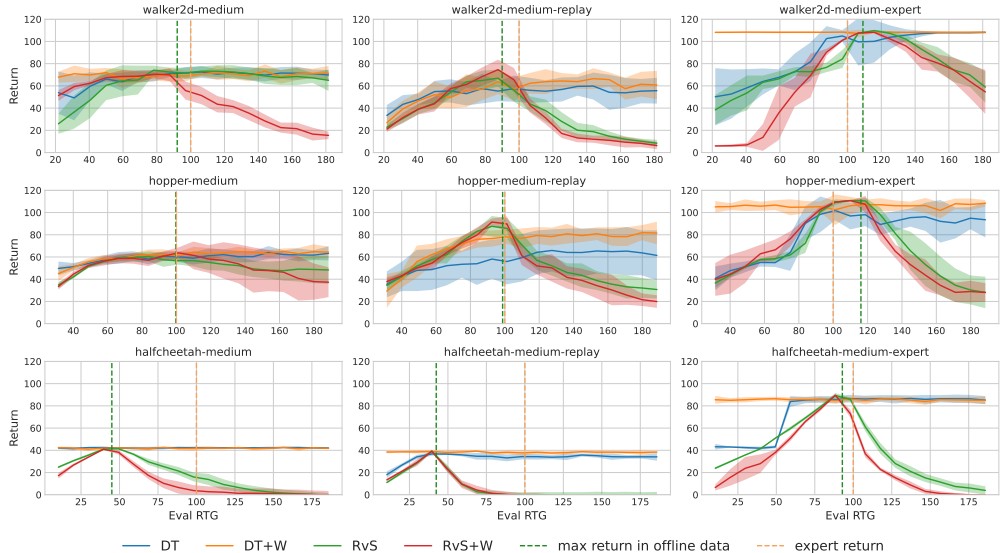

Figure 2: Performance of RvS and DT when conditioning on different evaluation RTGs. We report the mean and standard deviation of 10 seeds.

For each task, we consider the v2 `medium`, `med-replay` and `med-expert` offline datasets. The `medium` dataset contains 1M samples from a policy trained to approximately $\frac{1}{3}$ the performance of an expert policy. The `med-replay` dataset uses the replay buffer of a policy trained up to the performance of a `medium` policy. The `med-expert` dataset contains 1M samples generated by a `medium` policy and 1M samples generated by an expert policy.

**Baselines** We apply trajectory weighting to RvS (Emmons et al., 2021) and DT (Chen et al., 2021), two state-of-the-art BC methods. We compare their performance when trained on the original distribution and on the transformed distribution induced by our trajectory weighting (denoted as +W).

**Hyperparameters** For all datasets, we use $B = 20$ and $\lambda = 0.01$, and we set the temperature parameter $\kappa$ to be the difference between the highest return and the 90-th percentile: $\hat{r}^\star - \hat{r}_{90}$, whose value varies across the datasets. At test time, we set the evaluation RTG to be the expert return for each environment. The model architecture and the other hyperparamters are identical to what were used in the original paper. We provide a complete list of hyperparameters in Appendix B.2 and additional ablation experiments on $\lambda$ and $\kappa$ in Appendix C.

**Results** Table 1 shows the performance of RvS and DT and their variants. DT+W outperforms the original DT in $6/9$ datasets, achieving an average improvement of $8\%$. The improvement is significant in low-quality datasets (`med-replay`), which agrees with our analysis. Unlike in DT, trajectory weighting in RvS has varying effects, and the average performance of RvS+W is not better than that of RvS. To better understand this, we plot the achieved returns of RvS and DT when conditioning on different values of RTG. Figure 2 shows an interesting difference between behaviors of DT and RvS. DT is insensitive to the conditioning RTG, and continues performing stably even when conditioning on out-of-distribution RTGs. In contrast, the performance of RvS highly correlates with the evaluation RTG, but degrades quickly after a certain threshold. The performance crash problem of RvS shadows the improvement made by trajectory weighting.

## 4.2 RELIABLE EVALUATION VIA CONSERVATISM

The results in Section 4.1.2 introduce another challenging problem for return-conditioned BC in offline RL: generalization to out-of-distribution (ood) returns. While strong generalization beyond the offline dataset remains an ongoing challenge for the offline RL community (Wang et al., 2020; Zanette, 2021; Foster et al., 2021), we require the policy to be reliable and at least stay close to the data distribution to avoid catastrophic failure when conditioned on ood returns. In other words, we want the policy to be *conservative*. Figure 2 shows that DT enjoys self-conservatism, while RvS does

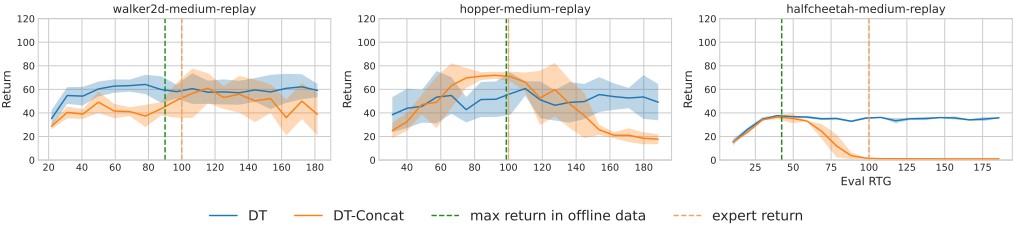

Figure 3: Performance of DT when the state and RTG tokens are concatenated. We report the mean and standard deviation of 10 seeds.

---

**Algorithm 2:** ConserWeightive Behavioral Cloning (CWBC) for RvS

1 **Input:** dataset $\mathcal{T}_{\text{offline}}$, number of iterations $I$, batch size $S$, regularization coefficient $\alpha$, initial parameters $\theta_0$

2 **for** *iteration* $i = 1, \ldots, I$ **do**

3      Sample a batch of trajectories $\mathcal{B} \leftarrow \{\tau^{(1)}, \ldots, \tau^{(S)}\}$ from $\mathcal{T}_{\text{offline}}$ using Algorithm 1.

4      **for** *every sampled trajectory* $\tau^{(i)}$ **do**

5          Samplie noise $\varepsilon$ as described in Section 4.2.1.

6          Compute noisy RTGs: $g_t^\varepsilon \leftarrow g_t + \varepsilon, 1 \leqslant t \leqslant |\tau^{(i)}|$.

         `// loss and regularizer defined in Equation (4) and (7)`

7      Perform gradient update of $\theta$ by minimizing the regularized empirical risk $\widehat{\mathcal{L}}_{\text{RvS}}^{\mathcal{B}}(\theta) + \alpha \cdot \widehat{\mathcal{C}}_{\text{RvS}}^{\mathcal{B}}(\theta)$.

8 **Output:** $\pi_\theta$

---

not. We hypothesize that the conservative behavior of DT comes from the transformer architecture. As the policy conditions on a sequence of both state tokens and RTG tokens to predict next action, the attention layers can choose to ignore the ood RTG tokens while still obtaining a good prediction loss. To test this hypothesis, we experiment with a slightly modified version of DT, where we concatenate the state and RTG at each timestep instead of treating them as separate tokens. By doing this, the model cannot ignore the RTG information in the sequence. We call this version DT-Concat. Figure 3 shows that the performance of DT-Concat is strongly correlated with the conditioning RTG, and degrades quickly when the target return is out-of-distribution. This result confirms our hypothesis.

However, conservatism does not have to come from the architecture, but can also emerge from a proper objective function, as commonly done in conservative value-based methods (Kumar et al., 2020; Fujimoto & Gu, 2021). In this section, we propose a novel conservative regularization for BC that explicitly encourages the policy to stay close to the data distribution. The intuition is to enforce the predicted actions when conditioning on large ood returns to stay close to the in-distribution actions. To do that, for a trajectory $\tau$ with high return, we inject *positive* random noise $\varepsilon \sim \mathcal{E}_\tau$ to its RTGs, and penalize the $\ell_2$ distance between the predicted action and the ground truth. Specifically, to guarantee we generate large ood returns, we choose a noise distribution $\mathcal{E}$ such that the perturbed initial RTG $g_1 + \varepsilon$ is at least $\widehat{r}^\star$, the highest return in the dataset. The next subsections instantiate the conservative regularizer in the context of RvS, and empirically evaluate its performance.

### 4.2.1 IMPLEMENTATION DETAILS

We apply conservative regularization to trajectories whose returns are above $\widehat{r}_q$, the $q$-th percentile of returns in the dataset. This makes sure that when conditioned on ood returns, the policy behaves similarly to high-return trajectories and not to a random trajectory in the dataset. We sample a scalar noise $\varepsilon \sim \mathcal{E}_\tau$ and offset the RTG of $\tau$ at every timestep by $\varepsilon$: $g_t^\varepsilon = g_t + \varepsilon, t = 1, \ldots, |\tau|$, resulting in the conservative regularizer:

$$\mathcal{C}_{\text{RvS}}(\theta) = \mathbb{E}_{\tau \sim \mathcal{T}, \, \varepsilon \sim \mathcal{E}_\tau} \left[ \mathbb{1}_{r_\tau > \widehat{r}_q} \cdot \frac{1}{|\tau|} \sum_{t=1}^{|\tau|} \left( a_t - \pi_\theta(s_t, \omega_t^\varepsilon) \right)^2 \right], \tag{7}$$

where $\omega_t^\varepsilon = (g_t + \varepsilon)/(H - t + 1)$ (cf. Equation (3)) is the noisy average RTG at timestep $t$. We observe that using the 95-th percentile of $\widehat{r}_{95}$ generally works well across different environments and datasets.

Table 2: Comparison of the normalized return on the D4RL locomotion benchmark. For BC and TD3+BC, we get the numbers from (Emmons et al., 2021). For IQL, we get the numbers from (Kostrikov et al., 2021b). For TTO, we get the numbers from (Janner et al., 2021). The results are averaged over 10 seeds.

| | RvS | RvS+W | RvS+C | RvS+W+C | DT | BC | TD3+BC | CQL | IQL | TTO |
|---|---|---|---|---|---|---|---|---|---|---|
| walker2d-medium | $73.3 \pm 5.7$ | $54.5 \pm 7.7$ | $71.3 \pm 4.9$ | $73.6 \pm 5.4$ | $71.5 \pm 3.9$ | 75.3 | 83.7 | 82.9 | 78.3 | $81.3 \pm 8.0$ |
| walker2d-med-replay | $54.0 \pm 12.1$ | $61.2 \pm 14.7$ | $62.0 \pm 13.5$ | $72.8 \pm 7.5$ | $53.4 \pm 12.2$ | 26.0 | 81.8 | 86.1 | 73.9 | $79.4 \pm 12.8$ |
| walker2d-med-expert | $102.2 \pm 2.3$ | $104.1 \pm 0.5$ | $102.1 \pm 10.2$ | $107.6 \pm 0.5$ | $99.8 \pm 21.3$ | 107.5 | 110.1 | 109.5 | 109.6 | $91.0 \pm 10.8$ |
| hopper-medium | $56.6 \pm 5.5$ | $62.5 \pm 7.1$ | $61.0 \pm 5.3$ | $62.9 \pm 3.6$ | $59.9 \pm 4.9$ | 52.9 | 59.3 | 64.6 | 66.3 | $67.4 \pm 11.3$ |
| hopper-med-replay | $87.7 \pm 9.7$ | $92.4 \pm 6.1$ | $91.5 \pm 3.5$ | $87.7 \pm 4.2$ | $56.4 \pm 20.1$ | 18.1 | 60.9 | 97.8 | 94.7 | $99.4 \pm 12.6$ |
| hopper-med-expert | $108.8 \pm 0.9$ | $108.4 \pm 1.8$ | $101.0 \pm 13.4$ | $110.0 \pm 2.8$ | $95.4 \pm 11.3$ | 52.5 | 98.0 | 102.0 | 91.5 | $106.0 \pm 1.1$ |
| halfcheetah-medium | $16.2 \pm 4.5$ | $4.0 \pm 5.4$ | $40.7 \pm 1.0$ | $42.2 \pm 0.7$ | $42.5 \pm 0.6$ | 42.6 | 48.3 | 49.1 | 47.4 | $44.0 \pm 1.2$ |
| halfcheetah-med-replay | $-0.4 \pm 2.7$ | $-0.8 \pm 2.2$ | $36.8 \pm 1.5$ | $40.4 \pm 0.8$ | $34.5 \pm 4.2$ | 36.6 | 44.6 | 47.3 | 44.2 | $44.1 \pm 3.5$ |
| halfcheetah-med-expert | $83.4 \pm 2.1$ | $69.1 \pm 3.7$ | $91.2 \pm 1.0$ | $91.1 \pm 2.0$ | $87.2 \pm 2.7$ | 55.2 | 90.7 | 85.8 | 86.7 | $40.8 \pm 8.7$ |
| # wins | / | 4 | 6 | 9 | / | / | / | / | / | |
| average | 64.6 | 61.7 | 73.1 | 76.5 | 66.73 | 51.86 | 75.3 | 80.6 | 77.0 | 72.6 |

We use the noise distribution $\mathcal{E}_\tau = \mathrm{Uniform}[l_\tau, u_\tau]$, where the lower bound $l_\tau = \hat{r}^\star - r_\tau$ so that the perturbed initial RTG $g_1^\varepsilon = r_\tau + \varepsilon$ is no less than $\hat{r}^\star$, and the upper bound $u_\tau = \hat{r}^\star - r_\tau + \sqrt{12\sigma^2}$ so that the standard deviation of $\mathcal{E}_\tau$ is equal to $\sigma$. We emphasize our conservative regularizer is distinct from the other conservative components proposed for the value-based offline RL methods. While those usually try to regularize the value function estimation to prevent extrapolation error (Fujimoto et al., 2019), we perturb the returns to generate ood conditioning and regularize the predicted actions.

When the conservative regularizer is used, the final objective for training RvS is $\mathcal{L}_{\mathrm{RvS}}(\theta) + \alpha \cdot \mathcal{C}_{\mathrm{RvS}}(\theta)$, in which $\alpha$ is the regularization coefficient. When trajectory reweighting is used in conjunction with the conservative regularizer, we obtain *ConserWeightive Behavioral Cloning (CWBC)*, which combines the best of both components. We provide a pseudo code for CWBC in Algorithm 2.

### 4.2.2 EMPIRICAL RESULTS

**Dataset** We evaluate the effectiveness of the conservative regularizer, as well as the performance of CWBC as a whole on the D4RL datasets (Fu et al., 2020) for the gym locomotion tasks.

**Baselines** We apply the conservative regularizer, which we denote as +C, to both RvS and RvS+W. In addition, we report the performance of three value-based methods: TD3+BC (Fujimoto & Gu, 2021), CQL (Kumar et al., 2020), and IQL (Kostrikov et al., 2021b) as a reference.

**Hyperparameters** We apply our conservatism regularization to trajectories whose returns are above the $q = 95$-th percentile return in the dataset, and perturb their RTGs as described in Section 4.2.1. We use a regularization coefficient of $\alpha = 1$. The evaluation protocol is similar to Section 4.1.2.

**Results** Table 2 reports the performance of different methods we consider. Our proposed framework CWBC with all components enabled (RvS+W+C) significantly outperforms the original RvS on $9/9$ datasets, with an average improvement of $18\%$ over RvS. RvS+W+C is also the best performing BC method in the table, and is competitive with the value-based methods. Conservative regularization consistently improves the results for both RvS and RvS+W. Trajectory weighting on its own can have varying effects on performance, but is synergistic when combined with RvS+C leading to our best performing model in RvS+W+C.

To better understand the impact of each component, we plot the achieved returns of RvS and other variants when conditioning on different values of conditioned RTG. Figure 4 shows that RvS generalizes poorly to out-of-distribution RTGs, which leads to significant performance drop when the evaluation RTG is larger than the best return in the dataset. Figure 4 illustrates the significant importance of encouraging conservatism for RvS, where RvS+C has much more stable performance, even when the evaluation RTG is $2\times$ the expert return. By explicitly asking the model to stay close to the data distribution, we achieve more reliable out-of-distribution performance, and avoid the performance crash problem. This leads to absolute performance improvement of RvS+C in Table 2. CWBC combines the best of both weighting and conservatism, which enjoys good performance when conditioning on high RTG values, as well as better robustness to large, out-of-distribution RTGs.

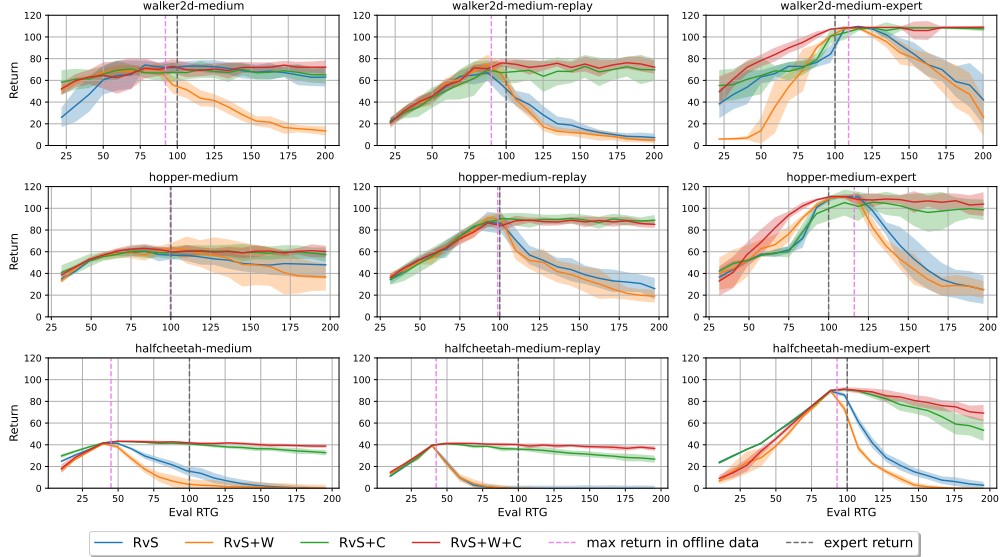

Figure 4: Performance of RvS and its variants when conditioning on different evaluation RTGs. We report the mean and standard deviation of 10 seeds.

In addition to the main results, we include ablations for different choices of conservative percentile $q$ and regularization coefficient $\alpha$ in Appendix C. Finally, we also evaluate CWBC in two more benchmarks: Atari games (Bellemare et al., 2013) and the D4RL Antmaze datasets. We present these results in Appendix D and E respectively.

## 5 CONCLUSION

We proposed ConserWeightive Behavioral Cloning (CWBC), a new framework that extends BC for offline RL with two novel components: trajectory weighting and conservative regularization. Trajectory weighting balances the bias-variance tradeoff that arises in learning from a suboptimal dataset, improving the performance of both DT and RvS. Next, we showed that while DT is self-conservative due to its attention architecture, we can recover this desired behavior even for RvS using our proposed conservative regularizer. Confirmed by the experiments, CWBC significantly improves the performance and stability of RvS.

While we made good progress for BC, advanced value-based methods such as CQL and IQL are still ahead and we believe further understanding of the tradeoffs in both kinds of approaches is important future work. Another promising direction from a data perspective is how to combine datasets from multiple environments to obtain diverse, high-quality data. Recent works have shown promising results in this direction (Reed et al., 2022). Last but not least, while CWBC significantly improves the performance and reliability of RvS, it is not able to extrapolate beyond the offline dataset. How to obtain extrapolation, or whether it is possible, is still an open question, and poses a persistent research opportunity for not only CWBC but the whole offline RL community.

## REPRODUCIBILITY STATEMENT

We present the practical implementation of our framework in Section 4.1.1 and Section 4. We include the implementation details of our paper in Appendix B, which contains information about the datasets we use, the open sourced code we base on, and the list of hyperparameters we use to reproduce our results. Finally, we submitted the source code in the supplementary material.

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

## A    LIST OF SYMBOLS

Table 3: Important symbols used in this paper.

| Symbol | Meaning | Definition |
|--------|---------|------------|
| $\mathcal{S}$ | state space | |
| $\mathcal{A}$ | action space | |
| $\tau$ | trajectory | |
| $|\tau|$ | trajectory length | |
| $\mathcal{T}$ | distribution of trajectories | |
| $\mathcal{T}_{\text{offline}}$ | offline dataset | |
| $\pi$ | policy | |
| $\theta$ | policy parameters | |
| $s_t$ | state at timestep $t$ | |
| $a_t$ | action at timestep $t$ | |
| $r_t$ | reward at timestep $t$ | |
| $r_\tau$ | trajectory return | $\sum_{t=1}^{|\tau|} r_t$ |
| $g_t$ | return-to-go at timestep $t$ | $\sum_{t'=t}^{|\tau|} r'_t$ |
| $H$ | maximum trajectory length | |
| $\omega_t$ | average return-to-go at timestep $t$ | $g_t/(H-t+1)$ |
| $\mathcal{L}_{\text{RvS}}$ | empirical risk of RvS | Equation (4) |
| $\mathcal{C}_{\text{RvS}}$ | conservative regularizer for RvS | Equation (7) |
| $f_\mathcal{T}(\tau)$ | probability density of trajectory $\tau \sim \mathcal{T}$ | |
| $p_{\tilde{\mathcal{T}}}(\tau)$ | probability density of trajectory $\tau \sim \tilde{\mathcal{T}}$ | Equation (5) |
| $b$ | index of a bin of trajectories in the offline dataset | |
| $|b|$ | size of bin $b$ | |
| $f_{\mathcal{T}_{\text{offline}}}(b)$ | proportion of trajectories in bin $b$ | $|b|/|\mathcal{T}_{\text{offline}}|$ |
| $\mathbb{P}_{\text{bin}}(b)$ | probability that bin $b$ is sampled | Equation (6) |
| $\bar{r}_\tau^b$ | average return of trajectories in bin $b$ | |
| $\hat{r}^\star$ | highest return in the offline dataset | |
| $\hat{r}_q$ | $q$-th percentile of the returns in the offline dataset | |

## B    IMPLEMENTATION DETAILS

### B.1    DATASETS AND SOURCE CODE

We train and evaluate our models on the D4RL (Fu et al., 2020) and Atari (Agarwal et al., 2020) benchmarks, which are available at `https://github.com/rail-berkeley/d4rl` and `https://research.google/tools/datasets/dqn-replay`, respectively. Our code-base is largely based on the RvS (Emmons et al., 2021) official implementation at `https://github.com/scottemmons/rvs`, and DT (Chen et al., 2021) official implementation at `https://github.com/kzl/decision-transformer`.

## B.2 DEFAULT HYPERPARAMETERS

Table 4: Hyperparameters used for locomotion experiments.

|  | Hyperparameter | Value |
|---|---|---|
| Model | Context length $K$ (DT) | 20 |
|  | Number of attention heads (DT) | 1 |
|  | Hidden layers | 2 for RvS, 3 for DT |
|  | Hidden dimension | 1024 for RvS, 128 for DT |
|  | Activation function | ReLU |
|  | Dropout | 0.0 for RvS, 0.1 for DT |
| Conservative regularizer | Conservative percentile $q$ | 95 |
|  | Noise standard deviation $\sigma$ | 1000 |
|  | Regularization coefficient $\alpha$ | 1.0 |
| Trajectory weighting | # bins $B$ | 20 |
|  | Smoothing parameter $\lambda$ | 0.01 |
|  | Smoothing parameter $\kappa$ | $\widehat{r}^{\star} - \widehat{r}_{90}$ |
| Optimization | Batch size | 64 |
|  | Learning rate | $1e-3$ for RvS, $1e-4$ for DT |
|  | Weight decay | $1e-4$ |
|  | Training iterations | 100000 |
| Evaluation | Target return | $1\times$ Expert return |

Table 5: Hyperparameters used for Atari experiments.

|  | Hyperparameter | Value |
|---|---|---|
| Model | Encoder channels | 32, 32, 64 |
|  | Encoder filter sizes | $8 \times 8, 4 \times 4, 3 \times 3$ |
|  | Encoder strides | 4, 2, 1 |
|  | Hidden layers | 4 |
|  | Hidden dimension | 1024 |
|  | Activation function | ReLU |
|  | Dropout | 0.1 |
| Conservative regularization | Conservative percentile $q$ | 95 |
|  | Noise std $\sigma$ | 50 for Breakout, Pong
500 for Qbert, Seaquest |
|  | Conservative weight $\alpha$ | 0.1 |
| Trajectory weighting | # bins $B$ | 20 |
|  | $\lambda$ | 0.1 |
|  | $\kappa$ | $\widehat{r}^{\star} - \widehat{r}_{50}$ |
| Optimization | Batch size | 128 |
|  | Learning rate | $6e-4$ |
|  | Weight decay | $1e-4$ |
|  | Training iterations | 25000 |
| Evaluation | Target return | 90 for Breakout ($1\times$ max in dataset)
2500 for Qbert ($5\times$ max in dataset)
20 for Pong ($1\times$ max in dataset)
1450 for Seaquest ($5\times$ max in dataset) |

# C ABLATION ANALYSIS

In this section, we investigate the impact of each of those hyperparameters on CWBC to give insights on what values work well in practice. We use the `walker2d` environment and the three related datasets for illustration. In all the experiments, when we vary one hyperparameter, the other hyperparameters are kept as in Table 4.

## C.1 TRAJECTORY WEIGHTING: SMOOTHING PARAMETERS $\lambda$ AND $\kappa$

Two hyperparameters $\kappa$ and $\lambda$ in Equation (6) affect the probability a bin index $b$ is sampled:

$$\mathbb{P}_{\text{bin}}(b) \propto \frac{f_{\mathcal{T}_{\text{offline}}}(b)}{f_{\mathcal{T}_{\text{offline}}}(b) + \lambda} \cdot \exp\Big(-\frac{|\bar{r}_\tau^b - \widehat{r}^\star|}{\kappa}\Big).$$

In practice, we have observed that the performance of CWBC is considerably robust to a wide range of values of $\kappa$ and $\lambda$.

**The impact of $\kappa$** The smoothing parameter $\kappa$ controls how we weight the trajectories based on their relative returns. Intuitively, smaller $\kappa$ gives more weights to high-return bins (and thus their trajectories), and larger $\kappa$ makes the transformed distribution more uniform. We illustrate the effect of $\kappa$ on the transformed distribution and the performance of CWBC in Figure 5. As in Section 4.1.2, we set $\kappa$ to be the difference between the empirical highest return $\widehat{r}^\star$ and the $z$-th percentile return in the dataset: $\kappa = \widehat{r}^\star - \widehat{r}_z$, and we vary the values of $z$. This allows the actual value of $\kappa$ to adapt to different datasets.

Figure 5 shows the results. The top row plots the distributions of returns before and after trajectory weighting for varying values of $\kappa$. We tested four values $z \in \{99, 90, 50, 0\}$, which correspond to four increasing values of $\kappa$. We mark the actual values of $\kappa$ in each dataset in the top tow[1]. For each dataset, we can see the transformed distribution using small $\kappa$ (orange) highly concentrates on high returns; as $\kappa$ increases, the density for low returns increases and the distribution becomes more and more uniform. The bottom row plots the corresponding performance of CWBC with different choices of $\kappa$. We select RvS+C as our baseline model, which does not have trajectory weighting but has the conservative regularization enabled. We can see that relatively small values of $\kappa$ (based on $\widehat{r}_{99}$, $\widehat{r}_{90}$ and $\widehat{r}_{50}$) perform well on all the three datasets, whereas large values (based on $\widehat{r}_0$) hurt the performance for the `med-expert` dataset, and even underperform the baseline RvS+C.

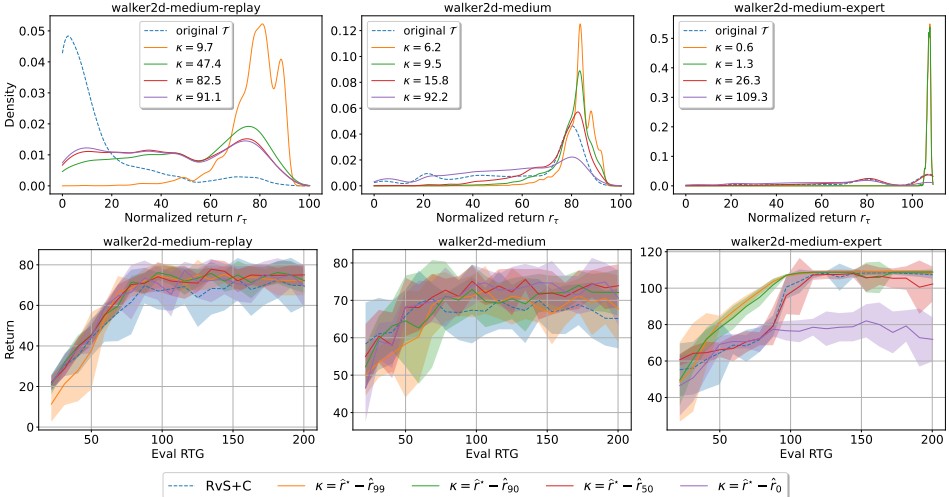

Figure 5: The influence of $\kappa$ on the transformed distribution (top) and on the performance of CWBC (bottom). The legend in each panel (top) shows the absolute values of $\kappa$ for easier comparison. In the bottom row, we also plot the results of RvS+C (no trajectory weighting) as a baseline.

---

[1] $\widehat{r}_0$ is defined to be the lowest return in the dataset: $\widehat{r}_0 = \min_{\tau \in \mathcal{T}_{\text{offline}}} r_\tau$.

**The impact of** $\lambda$    To better understand the role of $\lambda$, we can rewrite Equation (6) as

$$\mathbb{P}_{\text{bin}}(b) \propto \overbrace{f_{\mathcal{T}_{\text{offline}}}(b) \exp\big(-\tfrac{|\bar{r}_\tau^b - \hat{r}^\star|}{\kappa}\big)}^{\text{T1}} \cdot \overbrace{\big\{1/\big(f_{\mathcal{T}_{\text{offline}}}(b) + \lambda\big)\big\}}^{\text{T2}}.$$

Clearly, only T2 depends on $\lambda$. When $\lambda = 0$, T2 is canceled out and the above equation reduces to

$$\mathbb{P}_{\text{bin}}(b) \propto \exp\big(-\tfrac{|\bar{r}_\tau^b - \hat{r}^\star|}{\kappa}\big),$$

which purely depends on the relative return. As $\lambda$ increases, T2 is less sensitive to $f_{\mathcal{T}_{\text{offline}}}(b)$, and finally becomes the same for every $b \in [B]$ as $\lambda \to \infty$. In that scenario, $\mathbb{P}_{\text{bin}}(b)$ only depends on T1, which is the original frequency $f_{\mathcal{T}_{\text{offline}}}(b)$ weighted by the relative return.

The top row of Figure 6 plots the distributions of returns before and after trajectory weighting with different values of $\lambda$. When $\lambda = 0$, the distributions concentrate on high returns. As $\lambda$ increases, the distributions are more correlated with the original one, but still weights more on the high-return region compared to the original distribution due to the exponential term in T1. The bottom row of Figure 6 plots the actual performance of CWBC as $\lambda$ varies. All values of $\lambda$ produce similar results, which are consistently better than or comparable to training on the original datset (RvS+C).

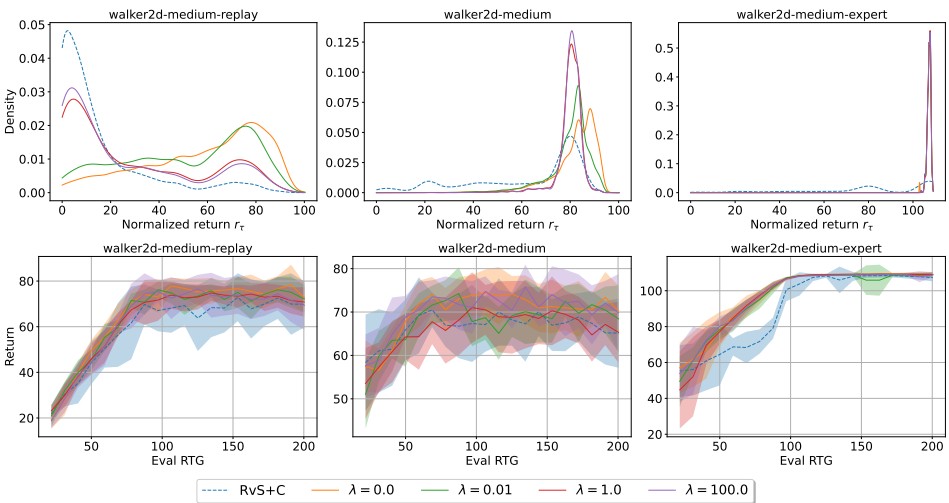

Figure 6: The influence of $\lambda$ on the transformed distribution (top) and on the performance of CWBC (bottom). We plot the result of RvS+C as the baseline.

## C.2    CONSERVATIVE REGULARIZATION: PERCENTILE $q$

We only apply the conservative regularization to trajectories whose return is above the $q$-th percentile of the returns in the dataset. Intuitively, a larger $q$ applies the regularization to fewer trajectories. We test four values for $q$: $0, 50, 95$, and $99$. For $q = 0$, our regularization applies to all the trajectories in the dataset. Figure 7 demonstrates the impact of $q$ on the performance of CWBC. $q = 95$ and $q = 99$ perform well on all the three datasets, while $q = 50$ and $q = 0$ lead to poor results for the `med-replay` dataset. This is because, when the regularization applies to trajectories of low returns, the regularizer will force the policy conditioned on out-of-distribution RTGs to stay close to the actions from low return trajectories. Since the `med-replay` dataset contains many low return trajectories (see Figure 5), such regularization results in poor performance. In contrast, `medium` and `med-expert` datasets contain a much larger portion of high return trajectories, and they are less sensitive to the choice of $q$.

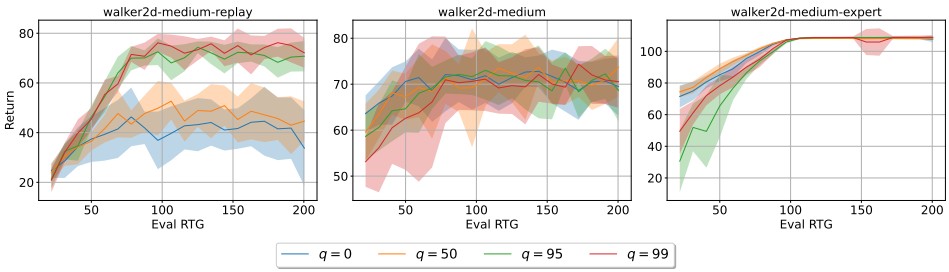

Figure 7: Performance of CWBC with different values of the conservative percentile $q$.

## C.3 REGULARIZATION COEFFICIENT $\alpha$

The hyperparameter $\alpha$ controls the weight of the conservative regularization in the final objective function of CWBC $\mathcal{L}_{\text{RvS}} + \alpha \cdot \mathcal{C}_{\text{RvS}}$. We show the performance of CWBC with different values of $\alpha$ in Figure 8. Not using any regularization ($\alpha = 0$) suffers from the performance crash problem, while overly aggressive regularization ($\alpha = 10$) also hurts the performance. CWBC is robust to the other non-extreme values of $\alpha$.

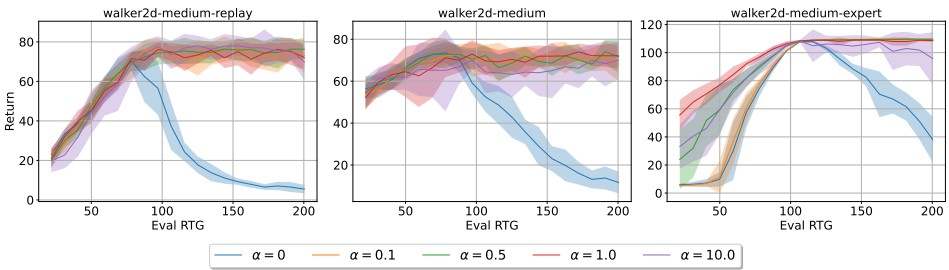

Figure 8: Performance of CWBC with different values of $\alpha$.

## D ADDITIONAL RESULTS ON ATARI GAMES

In addition to D4RL, we consider 4 games from the Atari benchmark (Bellemare et al., 2013): Breakout, Qbert, Pong, and Seaquest. Similar to (Chen et al., 2021), for each game, we train our method on 500000 transitions sampled from the DQN-replay dataset, which consists of 50 million transitions of an online DQN agent (Mnih et al., 2015). Due to the varying performance of the DQN agent in different games, the quality of the datasets also varies. While Breakout and Pong datasets are high-quality with many expert transitions, Qbert and Seaquest datasets are highly suboptimal.

**Hyperparameters** For trajectory weighting, we use $B = 20$ bins, $\lambda = 0.1$, and $\kappa = \widehat{r}^\star - \widehat{r}_{50}$. We apply conservative regularization with coefficient $\alpha = 0.1$ to trajectories whose returns are above $\widehat{r}_{95}$. The standard deviation of the noise distribution varies across datasets, as each different games have very different return ranges. During evaluation, we set the target return to $5 \times \widehat{r}^\star$ for Qbert and Seaquest, and to $1 \times \widehat{r}^\star$ for Breakout and Pong.

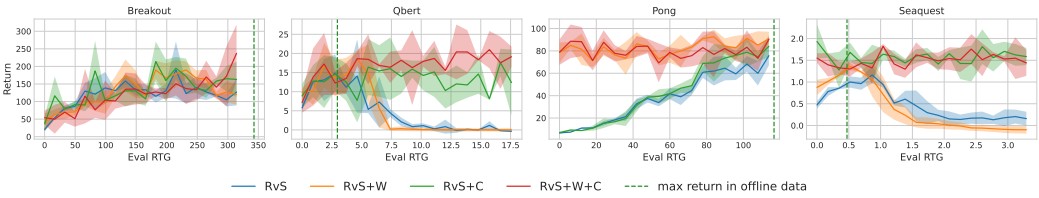

Figure 9: Performance of RvS and its variants on Atari games when conditioning on different evaluation RTGs.

Table 6: Comparison of the normalized return on Atari games. The results are averaged over 3 seeds. We include the results of DT, CQL, and BC from (Chen et al., 2021) for reference.

| | RvS | RvS+W | RvS+C | RvS+W+C | DT | CQL | BC |
|---|---|---|---|---|---|---|---|
| Breakout | $126.9 \pm 38.0$ | $120.1 \pm 28.43$ | $163.0 \pm 50.4$ | $237.3 \pm 82.1$ | $267.5 \pm 97.5$ | 211.1 | $138.9 \pm 61.7$ |
| Qbert | $-0.4 \pm 0.2$ | $0.0 \pm 0.4$ | $12.4 \pm 8.6$ | $19.1 \pm 2.7$ | $15.4 \pm 11.4$ | 104.2 | $17.3 \pm 14.7$ |
| Pong | $75.7 \pm 8.6$ | $90.7 \pm 6.4$ | $84.1 \pm 9.6$ | $90.4 \pm 1.9$ | $106.1 \pm 8.1$ | 111.9 | $85.2 \pm 20.0$ |
| Seaquest | $0.2 \pm 0.2$ | $-0.1 \pm 0.1$ | $1.6 \pm 0.2$ | $1.4 \pm 0.3$ | $2.5 \pm 0.4$ | 1.7 | $2.1 \pm 0.3$ |
| # wins | / | 1 | 4 | 4 | | | |
| average | 50.6 | 52.7 | 62.3 | 87.1 | 97.9 | 107.2 | 60.9 |

**Results** Table 6 summarizes the performance of RvS and its variants. CWBC (RvS+W+C) is the best method, outperforming the original RvS by 72% on average. Figure 9 clearly shows the effectiveness of the conservative regularization (+C). In two low-quality datasets Qbert and Seaquest, the performance of RvS degrades quickly when conditioning on out-of-distribution RTGs. By regularizing the policy to stay close to the data distribution, we achieve a much more stable performance. The trajectory weighting component (+W) alone has varying effects on performance because of the performance crash problem, but achieves state-of-the-art when used in conjunction with conservative regularization.

It is also worth noting that in both Qbert and Seaquest, CWBC achieves returns that are much higher than the best return in the offline dataset. This shows that while conservatism encourages the policy to stay close to the data distribution, it does not prohibit extrapolation. There is always a trade-off between optimizing the original supervised objective (which presumably allows extrapolation) and the conservative objective. This is very similar to other conservative regularizations used in value-based such as CQL or TD3+BC, where there is a trade-off between learning the value function and staying close to the data distribution.

## E    ADDITIONAL RESULTS ON D4RL ANTMAZE

Our proposed conservative regularization is especially important in dense reward environments such as gym locomotion tasks or Atari games, where choosing the target return during evaluation is a difficult problem. On the other hand, trajectory weighting is generally useful whenever the offline dataset contains both low-return and high-return trajectories. In this section, we consider Antmaze (Fu et al., 2020), a sparse reward environment in the D4RL benchmark to evaluate the generality of CWBC. Antmaze is a navigation domain in which the task is to control a complex 8-DoF "Ant" quadruped robot to reach a goal location. We consider 3 maze layouts: umaze, medium, and large, and 3 dataset flavors: v0, diverse, and play. We use the same set of hyperparameters as mentioned in B.2.

Table 7: Comparison of the success rate on the Antmaze environment. The results are averaged over 3 seeds. We include the results of DT, CQL, and BC from (Emmons et al., 2021) for reference.

| | RvS | RvS+W | RvS+C | RvS+W+C | DT | CQL | BC |
|---|---|---|---|---|---|---|---|
| umaze-v0 | $54.0 \pm 13.56$ | $65.0 \pm 18.03$ | $58.0 \pm 8.72$ | $65.0 \pm 12.85$ | 65.6 | 44.8 | 54.6 |
| umaze-diverse | $55.0 \pm 15.65$ | $46.0 \pm 16.85$ | $50.0 \pm 10.95$ | $42.0 \pm 7.48$ | 51.2 | 23.4 | 45.6 |
| medium-play | $0.0 \pm 0.0$ | $26.0 \pm 12.0$ | $0.0 \pm 0.0$ | $25.0 \pm 13.6$ | 1.0 | 0.0 | 0.0 |
| medium-diverse | $1.0 \pm 3.0$ | $24.0 \pm 15.62$ | $1.0 \pm 3.0$ | $23.0 \pm 11.0$ | 0.6 | 0.0 | 0.0 |
| large-play | $0.0 \pm 0.0$ | $4.0 \pm 4.9$ | $0.0 \pm 0.0$ | $5.0 \pm 6.71$ | 0.0 | 0.0 | 0.0 |
| large-diverse | $0.0 \pm 0.0$ | $10.0 \pm 10.0$ | $0.0 \pm 0.0$ | $17.0 \pm 11.87$ | 0.2 | 0.0 | 0.0 |
| # wins | / | 4 | 1 | 4 | | | |
| average | 18.3 | 29.2 | 18.2 | 29.5 | 19.8 | 11.4 | 16.7 |

**Results** Table 7 summarizes the results. As expected, the conservative regularization is not important in these tasks, as the target return is either 0 (fail) or 1 (success). However, the trajectory weighting significantly boosts performance, resulting in an average of 60% improvement over the original RvS.

## F TRAJECTORY WEIGHTING VERSUS HARD FILTERING

An alternative to trajectory weighting is hard filtering (+F), where we train the model on only top 10% trajectories with the highest returns. Filtering can be considered a hard weighting mechanism, wherein the transformed distribution only has support over trajectories with returns above a certain threshold.

### F.1 HARD FILTERING FOR RvS

When using hard filtering for RvS, we also consider combining it with the conservative regularization. Table 8 and Figure 10 compare the performance of trajectory weighting and hard filtering when applied to RvS. While RvS+F+C also gains notable improvements , it lags behind RvS+W+C and seems to erode the benefits of conservatism alone in RvS+C. This agrees with our analysis in Section 4.1. While hard filtering achieves the same effect of reducing bias, it completely removes the low-return trajectories, resulting in highly increased variance. Our trajectory weighting upweights the good trajectories but aims to stay close to the original data distribution, balancing this bias-variance tradeoff. This is clearly shown in Figure 10, where RvS+W+C has much smaller variance when conditioning on large RTGs.

Table 8: Comparison of trajectory weighting (+W) and hard filtering (+F) on D4RL locomotion benchmarks. The results are averaged over 10 seeds.

|  | RvS | RvS+W | RvS+C | RvS+W+C | RvS+F | RvS+F+C |
|---|---|---|---|---|---|---|
| walker2d-medium | $73.3 \pm 5.7$ | $54.5 \pm 7.7$ | $71.3 \pm 4.9$ | $73.6 \pm 5.4$ | $60.9 \pm 4.9$ | $68.2 \pm 7.1$ |
| walker2d-med-replay | $54.0 \pm 12.1$ | $61.2 \pm 14.7$ | $62.0 \pm 13.5$ | $72.8 \pm 7.5$ | $47.1 \pm 7.7$ | $53.9 \pm 11.0$ |
| walker2d-med-expert | $102.2 \pm 2.3$ | $104.1 \pm 0.5$ | $102.1 \pm 10.2$ | $107.6 \pm 0.5$ | $101.7 \pm 3.3$ | $105.4 \pm 0.6$ |
| hopper-medium | $56.6 \pm 5.5$ | $62.5 \pm 7.1$ | $61.0 \pm 5.3$ | $62.9 \pm 3.6$ | $62.4 \pm 5.0$ | $65.7 \pm 6.4$ |
| hopper-med-replay | $87.7 \pm 9.7$ | $92.4 \pm 6.1$ | $91.5 \pm 3.5$ | $87.7 \pm 4.2$ | $91.2 \pm 5.3$ | $92.1 \pm 2.9$ |
| hopper-med-expert | $108.8 \pm 0.9$ | $108.4 \pm 1.8$ | $101.0 \pm 13.4$ | $110.0 \pm 2.8$ | $97.5 \pm 15.0$ | $105.8 \pm 3.5$ |
| halfcheetah-medium | $16.2 \pm 4.5$ | $4.0 \pm 5.4$ | $40.7 \pm 1.0$ | $42.2 \pm 0.7$ | $1.4 \pm 3.3$ | $36.2 \pm 2.5$ |
| halfcheetah-med-replay | $-0.4 \pm 2.7$ | $-0.8 \pm 2.2$ | $36.8 \pm 1.5$ | $40.4 \pm 0.8$ | $-0.1 \pm 3.5$ | $35.7 \pm 2.8$ |
| halfcheetah-med-expert | $83.4 \pm 2.1$ | $69.1 \pm 3.7$ | $91.2 \pm 1.0$ | $91.1 \pm 2.0$ | $46.0 \pm 1.5$ | $83.2 \pm 5.0$ |
| # wins | / | 4 | 6 | 9 | 3 | 5 |
| average | 64.6 | 61.7 | 73.1 | 76.5 | 56.5 | 71.8 |

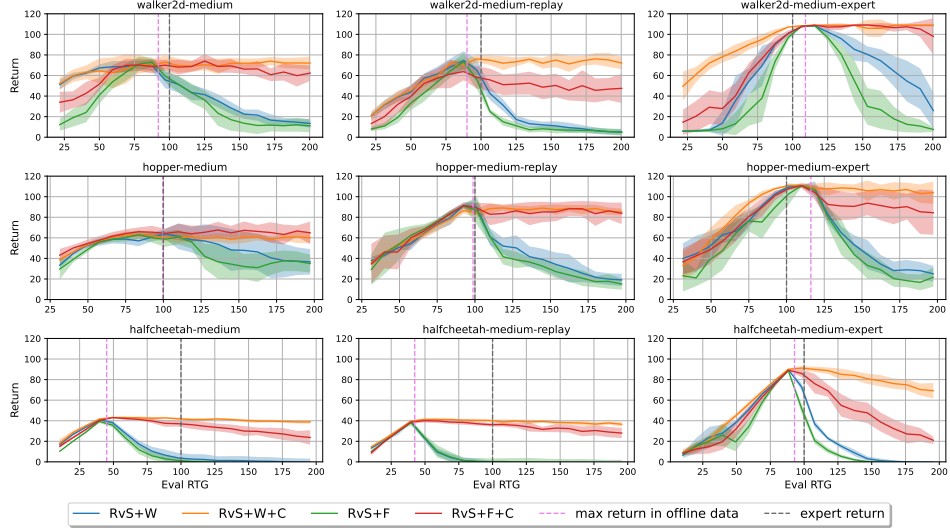

Figure 10: Comparison of trajectory weighting and hard filtering.

### F.2 HARD FILTERING FOR UNCONDITIONAL BC

Hard filtering can also be applied to ordinary BC. This is equivalent to Filtered BC in (Emmons et al., 2021). Table 9 compares Filtered BC and CWBC. CWBC performs comparably well in `medium` and `med-expert` datasets, and outperforms Filtered BC significantly with an average improvement of 12% in `med-replay` datasets. We believe that in low-quality datasets, even when we filter out 90% percent of the data, the quality of the remaining trajectories is still very diverse that simple imitation learning is not good enough. CWBC is able to learn from such diverse data, and by conditioning on expert return at test time, we can recover an efficient policy.

Table 9: The normalized return on D4RL for Filtered BC, RvS, and CWBC. For Filtered BC, we get the numbers from (Emmons et al., 2021).

|  | Filtered BC | RvS | RvS+W+C |
|---|---|---|---|
| walker2d-medium | 75.0 | $73.3 \pm 5.7$ | $73.6 \pm 5.4$ |
| hopper-medium | 56.9 | $56.6 \pm 5.5$ | $62.9 \pm 3.6$ |
| halfcheetah-medium | 42.5 | $16.2 \pm 4.5$ | $42.2 \pm 0.7$ |
| medium average | 58.1 | 48.7 | 59.6 |
| walker2d-med-replay | 62.5 | $54.0 \pm 12.1$ | $72.8 \pm 7.5$ |
| hopper-med-replay | 75.9 | $87.7 \pm 9.7$ | $87.7 \pm 5.2$ |
| halfcheetah-med-replay | 40.6 | $-0.4 \pm 2.7$ | $40.4 \pm 0.8$ |
| med-replay average | 59.7 | 47.1 | 67.0 |
| walker2d-med-expert | 109.0 | $102.2 \pm 2.3$ | $107.6 \pm 0.5$ |
| hopper-med-expert | 110.9 | $108.8 \pm 0.9$ | $110.0 \pm 2.8$ |
| halfcheetah-med-expert | 92.9± | $83.4 \pm 2.1$ | $91.1 \pm 2.0$ |
| med-expert average | 104.3 | 98.1 | 102.9 |
| average | 74.0 | 64.6 | 76.5 |

# G   BIAS-VARIANCE TRADEOFF ANALYSIS

We formalize our discussion on the bias-variance tradeoff when learning from a suboptimal distribution mentioned in Section 4.1. The objective functions for training DT (2) and RvS (4) can be rewritten as:

$$\min_\theta \mathcal{L}_{p_\mathcal{D}}(\theta) = \mathbb{E}_{\tau \sim \mathcal{T}}\left[D(\tau, \pi_\theta)\right] \tag{8}$$

$$= \mathbb{E}_{r \sim p_\mathcal{D}(r), \tau \sim \mathcal{T}_r}\left[D(\tau, \pi_\theta)\right]. \tag{9}$$

In which, $p_\mathcal{D}(r)$ is the data distribution over trajectory returns, $\mathcal{T}_r$ is a uniform distribution over the set of trajectories whose return is $r$, and $D(\tau, \pi_\theta)$ is the supervised loss function with respect to the sampled trajectory $\tau$. For DT, $D(\tau, \pi_\theta) = \frac{1}{|\tau|}\sum_{t=1}^{|\tau|}\left(a_t - \pi_\theta(g_{t-K:t}, s_{t-K:t}, a_{t-K:t-1})\right)^2$, and for RvS, $D(\tau, \pi_\theta) = \frac{1}{|\tau|}\sum_{t=1}^{|\tau|}\left(a_t - \pi_\theta(s_t, \omega_t)\right)^2$. Equation (9) is equivalent to first sampling a return $r$, then sampling a trajectory $\tau$ whose return is $r$, and calculating the loss on $\tau$. Ideally, we want to train the model from an optimal return distribution $p^\star(r)$, which is centered around the expert return $r^\star$:

$$\min_\theta \mathcal{L}_{p^\star}(\theta) = \mathbb{E}_{r \sim p^\star(r), \tau \sim \mathcal{T}_r}\left[D(\tau, \pi_\theta)\right]. \tag{10}$$

In practice, we only have access to the suboptimal return distribution $p_\mathcal{D}(r)$, which leads to a biased training objective with respect to $p^\star(r)$. While the dataset is fixed, we can transform the data distribution $p_\mathcal{D}(r)$ to $q(r)$ that better estimates the ideal distribution $p^\star(r)$. The objective function with respect to $q$ is:

$$\min_\theta \mathcal{L}_q(\theta) = \mathbb{E}_{r \sim q(r), \tau \sim \mathcal{T}_r}\left[D(\tau, \pi_\theta)\right] \tag{11}$$

$$= \mathbb{E}_{r \sim p_\mathcal{D}(r), \tau \sim \mathcal{T}_r}\left[\frac{q(r)}{p_\mathcal{D}(r)} \cdot D(\tau, \pi_\theta)\right] \tag{12}$$

In the extreme case, $q(r) = \mathbb{1}[r = r^\star]$, which means we only train the policy on trajectories whose return matches the expert return $r^\star$. However, since offline datasets often contain very few expert trajectories, this $q$ leads to a very high-variance training objective. An optimal distribution $q$ should lead to a training objective that balances the bias-variance tradeoff. We quantify this by measuring the $\ell_2$ of the difference between the gradient of $\mathcal{L}_q(\theta)$ and the gradient of the optimal objective function $\mathcal{L}_{p^\star}(\theta)$. Analogous to Kumar & Levine (2020), we can prove that for some constants $C_1, C_2, C_3$, with high confidence:

$$\mathbb{E}\left[||\nabla_\theta\mathcal{L}_q(\theta) - \nabla_\theta\mathcal{L}_{p^\star}(\theta)||_2^2\right] \leqslant C_1 \cdot \mathbb{E}_{r \sim q(r)}\left[\frac{1}{N_r}\right] + C_2 \cdot \frac{d_2(q||p_\mathcal{D})}{|\mathcal{D}|} + C_3 \cdot D_{\text{TV}}(p^\star, q)^2. \tag{13}$$

In which, $N_r$ is the number of trajectories in dataset $\mathcal{D}$ whose return is $r$, $d_2$ is the exponentiated Renyi divergence, and $D_{\text{TV}}$ is the total variation divergence. The right hand side of inequality (13) shows that an optimal distribution $q$ should be close to the data distribution $p_\mathcal{D}$ to reduce variance, while approximating well $p^\star$ to reduce bias. As shown in Kumar & Levine (2020), $q(r) \propto \frac{N_r}{N_r + K} \cdot \exp(-\frac{|r - r^\star|}{\kappa})$ minimizes this bound, which inspires our trajectory weighting.

