# OpenReview forum: "ConserWeightive Behavioral Cloning for Reliable Offline Reinforcement Learning"
_ICLR.cc/2023/Conference — Submitted to ICLR 2023_

### Official Review · Reviewer_uS7R · 2022-10-22

**Confidence:** 4
**Correctness:** 4
**Technical Novelty And Significance:** 2
**Empirical Novelty And Significance:** 2
**Recommendation:** 5

**Clarity, Quality, Novelty And Reproducibility:**

Clarity:
The description and analysis in this paper are relatively clear. I especially like Figure 1 and Figure 3, which greatly help the readers to understand the intuition behind trajectory-weighting and the effect of out-of-distribution rewards.

Quality:
Overall, the quality is okay but seems not good enough as a top-tier conference paper.

As for equation (5), there are two hyper-parameters introduced for trajectory weighting. A smaller value of \lambda or \kappa will give more weights for higher-return trajectories in the offline dataset. Then why do we need both of them? Can the trajectory-weighting approach be simplified by discarding one hyper-parameter? As mentioned in the paper, a similar equation was used for model-based optimization with optimality analysis. It will be great if the proposed equation (5) can also be built on a theoretical foundation to verify its correctness and usefulness for offline RL.

There are many hyper-parameters, \lambda and \kappa in equation (5), the number of bins B, the percentile value q, and the weight of C_{RvS} \alpha. When we change to other tasks, is it easy to tune the proposed method? According to Table 5 in Appendix, these hyper-parameters are changed for experiments on Atari environments. Is there any guidance about how to adjust these hyper-parameters when facing a new offline dataset?

Could you please add more metrics (e.g. https://arxiv.org/pdf/2108.13264.pdf) to show the benefit is statistically significant? Especially for the benefit of trajectory-weighting and reward perturbation separately, the improvement for each of these components seems marginal from Table1&2 considering the standard error.

Novelty:

The proposed method is novel in the context of improving DT and RvS. But the idea is not general enough to benefit more methods or inspire more research in the offline RL area. So the significance or impact of this work is not great.

Reproducibility:
There are source codes provided in the supplementary material and detailed hyper-parameters are listed in Appendix. So the reproducibility is okay.


**Strength And Weaknesses:**

Strengths:
This paper is generally well-written. The presentation is clear and easy to follow.
The proposed method is technically simple and easy to implement
There are extensive experiments on different environments, locomotion tasks, Atari games, and AntMaze (see Appendix). The comparison with hard-filtering BC in Appendix is a good supplement for the main text.

Weaknesses:
The paper is mainly considering some tricks to improve DT and RvS, which seems incremental. The proposed idea is not general and enlightening enough to inspire follow-up works.
The proposed methods are more like engineering techniques, instead of a principled approach with theoretical foundations or brilliant insight.
The comparison with baselines is not thorough. For example, the trajectory transformer is missing in Table 1&2. Table 6&7 do not have BC, IQL, etc. The proposed method fails to beat trajectory transformer as a SOTA method on locomotion tasks.


**Summary Of The Paper:**

This paper considers the offline RL problem and proposed two techniques (trajectory weighting, and reward perturbation) to improve Decision Transformer (DT) and Reinforcement Learning via Supervised Learning (RvS).
Specifically, trajectory weighting transforms the trajectory distribution of the original offline dataset. So the low-return trajectories are down-weighted in the new distribution while the target policy can still learn useful information from these low-return trajectories. Reward perturbation helps improve the performance of RvS when the target return-to-go is too high at test time. This approach is to perturb the return-to-go during RvS training, so that the learned policy yields actions from high-return trajectories when conditioned on out-of-distribution high return-to-go.
The experiments are conducted on the standard offline RL dataset, D4RL. On Hopper, HalfCheetah, and Walker2d, the trajectory-weighting technique turns out to be beneficial for both DT and RvS, and reward perturbation shows improvement for RvS.


**Summary Of The Review:**

To sum up, this paper proposes simple techniques to improve DT and RvS with extensive experiments. However, the scope is kind of limited, and the performance is not SOTA.

---

> ### Author Response · Authors · 2022-11-13
> **Author response**
>
> We thank the reviewer for the constructive feedback and for the recognition of the strengths of CWBC. We have updated the paper to include a formal discussion of the bias-variance tradeoff and more baseline results, which we marked in red. We answer each of the reviewer's concerns below.
>
> > The proposed idea is not general and enlightening enough to inspire follow-up works
>
> Our work aims to point out the challenges faced by existing return-conditioned methods and propose simple solutions for them. As these methods are becoming more popular and used across many domains, we believe a better understanding of their shortcomings is a good contribution to the community. Our proposed idea is general in the context of return-conditioned methods, and can be used to improve the performance of any existing models.
>
> > The comparison with baselines is not thorough and the proposed method does not beat SOTA
>
> As mentioned earlier, we want to show that our framework can be used to improve the performance of existing models, therefore we compared CWBC with the original models (DT and RvS), and also some SOTA value-based methods for reference (CQL, IQL, TD3+BC). We chose DT and RvS as the base methods because of their conceptual simplicity and competitive performance compared to trajectory transformers. As suggested by the reviewer, we have also added trajectory transformer in Table 2, and value-based methods in Table 6&7 for completeness. CWBC actually performs better than trajectory transformers on average in the mujoco benchmark, and is the best return-conditioned method. While CWBC does not beat CQL in this benchmark, which we have acknowledged in section 5, we believe this does not take away our contribution in narrowing the gap between value-based methods and conditional BC-based methods.
>
> > Why do we need both \lambda and \kappa
>
> We note that \lambda and \kappa have different roles in trajectory weighting. The intuition of equation (5) is that a trajectory is assigned more weights if its return has a high density in the original distribution (first term), and/or its return is high (second term). Therefore, \lambda controls how close the transformed distribution is to the original distribution, while \kappa controls how much we want to favor the high-return trajectories. If we remove one of them, we have less control over the shape of the transformed distribution. We perform an ablation study on these two hyperparameters in Appendix C.
>
> > Is it easy to tune the proposed method?
>
> In our experience, one set of hyperparameters can work well for all tasks in the same benchmark. For Atari, the only thing we needed to customize for each task was the noise standard deviation, because the return ranges of different games were significantly different. For Mujoco, as the return ranges were somewhat similar among tasks, we were able to just use one set of hyperparameters.
>
> > The improvement for each of these components seems marginal from Table1&2 considering the standard error.
>
> Table 1 shows that trajectory weighting (DT+W) improves the performance of DT by 8% on average. The improvement in RvS+W is not clear because of the performance crash problem we discussed in sections 4.1 and 4.2. Table 2 shows that conservative regularization alone (RvS+C) improves the performance of RvS by 13% on average, and trajectory weighting (RvS+W+C) gains an additional improvement of 5%. We believe these improvement gains are significant, even after considering the standard error. CWBC also generally enjoys a smaller variance in performance compared to the original model. Moreover, in Atari and Antmaze, CWBC outperforms the original model by 72% and 60%, respectively, which are significant gains in performance.

---

> > ### Comment · Reviewer_uS7R · 2022-12-02
> > **Thanks for the reply**
> >
> > I thank the authors for the reply. It partially addressed my questions, but I'm still concerned about the scope, significance, and novelty of the proposed method.
> >
> > In its current form, the proposed method is limited in the scope of return-based offline RL approaches (only DT and RvS as shown in this paper). Considering that return-based methods do not perform as well as value-based methods, the significance of the technique seems not great enough. The technique of trajectory reweighting (equation 5) is adapted from existing work, so it is not novel itself. Considering the high bar of acceptance in ICLR, I may keep my rating.

---

> > > ### Author Response · Authors · 2022-12-02
> > > **Author response**
> > >
> > > We thank the reviewer for the response. We address the reviewer's remaining concerns below.
> > > > The proposed method is limited in the scope of return-based offline RL approaches
> > >
> > > Return-conditioned RL is a major class of algorithms for offline RL which keeps growing in size and importance. Compared to the value-based approach, return-conditioned RL has significant potential and benefits:
> > > - Compared to value-based methods, they are much simpler and more stable to train.
> > > - This paradigm introduces a new view that connects offline RL with supervised learning and generative modeling, allowing us to utilize many exciting advances in these areas, such as diffusion models for more accurate long-horizon planning [2].
> > > - A major benefit of return-conditioned methods is the unprecedented scalability with respect to both data and model size. For example, Multi-game DT [4] was the first to achieve super-human performance in all Atari games using a single neural network, and its performance increases with respect to model scaling, whereas the other methods either saturate or have much slower performance growth.
> > > - Many recent publications at ML venues have been proposed to better understand return-conditioned RL by drawing connections to hindsight information matching [5], studying their exploration strategies in the online setting [6], and deriving theoretical guarantees [7, 8].
> > >
> > > Therefore we believe it’s important to obtain a better understanding of the challenges these methods face, and also how we can address these challenges efficiently. Our paper pointed out two main challenges: the bias-variance tradeoff due to learning from sub-optimal datasets and poor generalization to out-of-distribution returns. Our proposals have consistently improved the performance and reliability of 2 major existing methods in all benchmarks. Moreover, most of the previous works also only focused on one class of algorithm, which was either value-based or return-conditioned, so we do not understand why this is considered a limitation.
> > >
> > > > Considering that return-based methods do not perform as well as value-based methods, the significance of the technique seems not great enough
> > >
> > > We believe that just because one class of methods underperforms another class, it doesn't mean everyone should only work on the outperforming class. This promotes a monoculture of ideas and excessive reliance on SOTA rather than improved understanding. In fact, the unsatisfying empirical performance of the return-conditioned approach was the main motivation for us to study their challenges and propose solutions to improve their performance. And indeed, CWBC has significantly narrowed the gap between existing conditional BC methods and the SOTA value-based methods in all tasks we considered. Moreover, previous works such as DT and RvS did not outperform the value-based methods either, but they were accepted because of their great potential and benefits compared to the value-based approach.

---

> > > > ### Author Response · Authors · 2022-12-02
> > > > **Author response (continued)**
> > > >
> > > > > The technique of trajectory reweighting (equation 5) is adapted from existing work, so it is not novel itself
> > > >
> > > > MINs [1] was proposed for model-based optimization, while our work is proposed in the context of offline RL. In model-based optimization, the dataset consists of (x, y) pairs, whereas in offline RL we have a dataset of trajectories. Therefore, our trajectory weighting reweights the entire trajectories by their returns, as opposed to [1] which reweights individual data points. We believe that adopting a known technique from another domain, drawing connections, and empirically validating it in a new domain is an important contribution. Moreover, we also propose a novel conservative regularization that significantly improves the stability and reliability of RvS when conditioning on out-of-distribution returns. While conservatism has been studied extensively for the value-based approach, we are the first to study the notion and importance of conservatism in return-based methods.
> > > >
> > > > [1] Aviral Kumar and Sergey Levine. Model inversion networks for model-based optimization. Advances in Neural Information Processing Systems, 33:5126–5137, 2020.
> > > >
> > > > [2]  Janner, M., Du, Y., Tenenbaum, J. &amp; Levine, S.. (2022). Planning with Diffusion for Flexible Behavior Synthesis. Proceedings of the 39th International Conference on Machine Learning, in Proceedings of Machine Learning Research 162:9902-9915 Available from https://proceedings.mlr.press/v162/janner22a.html.
> > > >
> > > > [4] Lee, K. H., Nachum, O., Yang, M., Lee, L., Freeman, D., Xu, W., ... & Mordatch, I. (2022). Multi-Game Decision Transformers. Advances in neural information processing systems, 35
> > > >
> > > > [5] Furuta, H., Matsuo, Y., & Gu, S. S. (2021, September). Generalized Decision Transformer for Offline Hindsight Information Matching. In International Conference on Learning Representations.
> > > >
> > > > [6] Zheng, Q., Zhang, A. & Grover, A.. (2022). Online Decision Transformer. Proceedings of the 39th International Conference on Machine Learning, in Proceedings of Machine Learning Research 162:27042-27059 Available from https://proceedings.mlr.press/v162/zheng22c.html.
> > > >
> > > > [7] Kumar, A., Hong, J., Singh, A., & Levine, S. (2021, September). Should I Run Offline Reinforcement Learning or Behavioral Cloning?. In International Conference on Learning Representations.
> > > >
> > > > [8] Brandfonbrener, D., Bietti, A., Buckman, J., Laroche, R., & Bruna, J. (2022). When does return-conditioned supervised learning work for offline reinforcement learning?. Advances in neural information processing systems, 35

---

> ### Author Response · Authors · 2022-11-16
> **Checking in**
>
> Thank you again for your review. We hope you have had a chance to read our response. We believe our reply addresses the weaknesses and questions raised by this review. Since the discussion period is short and drawing to a close soon, are there further questions or concerns we should discuss? Thank you.

---

### Official Review · Reviewer_vF8h · 2022-10-24

**Confidence:** 4
**Correctness:** 3
**Technical Novelty And Significance:** 2
**Empirical Novelty And Significance:** 3
**Recommendation:** 6

**Clarity, Quality, Novelty And Reproducibility:**

The paper is easy to follow and the contributions are written clearly. The authors also provide code for the paper which seems easy to understand too. In terms of quality and significance of the work, I am not sure if this work can meet the standard since the contributions are rather not much novel. The paper identifies an important problem of return conditioning on lower quality datasets and the need for trajectory reweighting, but experimental gains are not that significant and such ideas have been considered in the past well.

**Strength And Weaknesses:**


	1. The paper tackles an interesting problem of return conditioning in offline RL, where even if there exists expert trajectories, the returns from offline dataset at test time can still be smaller compared to the expert data collecting policy. It argues that conditional BC can be improved to tackle this problem, by a simple mechanism of trajectory reweighting.
	2. Figure 1 well motivates the problem this paper tries to tackle. However, it would be helpful to understand more about this phenomenon - ie, to what extent does performance actually differ between the original and transformed distributions. This figure tries to argue that distribution of returns can be different depending on the type of data distribution. Existing benchmarks like D4RL considers expert/medium/medium-expert datasets, and it would be interesting to see the significance of this phenomenon in actual benchmark datasets used in practice.
	3. The core idea proposed in the paper is to consider a reweighted trajectory distribution that conditions more on the high return trajectories, such that conditioning BC on this distribution can improve performance. This leads to the problem of high variance though, since there may be fewer high return samples available.
	4. Equation 5 is a mechanism inspired from previous work for reweighting the trajectories. I think the main problem is the approximation of equation 5 in this case that is proposed. Moreover, equation 5 in its current form is not completely novel and is a modification of existing approaches that typically re-weights training data distribution as well;
	5. The mechanism is applied to existing baselines to show that performance can be marginally improved due to the addition of this reweighting mechanism. It is not well explained in the paper why such a form of regularization is needed, given the reweighting mechanism already being proposed. Are there simple demonstrations explaining the need for it? I ask this because regularization schemes in offline RL has been well studied in recent past, all on D4RL too - so unless a different form of dataset or other benchmarks are being used, this conservatism based update seems like a rather forced introduction to this paper?
The paper tries to well motivate the problem, but I am not sure if the experimental results presented in this form are fully convincing or not. I understand the limitations of existing offline RL experiments to d4rl benchmarks only, but recently there are new benchmarks such as v-d4rl as well, with visual observations, that might be worth considering to further make additions to experiments in this paper. Current results comparing to only two baselines seems rather limited. Results on the Atari domains, figure 9 shows marginal improvements only?

**Summary Of The Paper:**


This paper considers offline RL with return conditioning, similar to the recent line of works on decision transformers. It considers the problem of return conditioning especially when rewards can be arbitrary depending on the offline data distribution, even if considering expert trajectories. It considers two key methods to tackle arbitrary reward conditioning - trajectory weighting and conservative regularization. These ideas are proposed assuming that expert trajectories can still contain low and high return trajectories. Experimental results are claimed to achieve significant gains of 18% and 8% depending on the state of the art offline RL algorithm, conditioning on returns, being used.

**Summary Of The Review:**


The paper considers an important problem but the novelty, either algorithmic, or significance of experimental gains are rather limited. While the paper is easy to follow and argues for the need for such weighting - it seems like the two contributions are rather forced together to make this into a complete paper. It is not clear why conservative updates are still needed when trajectories are re-weighted, and the two additions to existing baselines are probably not the most important needed to tackle this problem.

I think this paper is a useful contribution, but perhaps not significantly novel for acceptance in a venue like ICLR.

I am however willing to re-consider my review and score if there is a more detailed justification of the proposed approach, and the importance of it, to improve existing performance on benchmarks.

---

> ### Author Response · Authors · 2022-11-13
> **Author response (Part 1 of 2)**
>
> We thank the reviewer for the constructive feedback and for the recognition of the strengths of CWBC. We have updated the paper to include a formal discussion of the bias-variance tradeoff and more baseline results, which we marked in red. We answer each of the reviewer's concerns below.
>
> > To what extent does performance actually differ between the original and transformed distributions
>
> In the Mujoco benchmark, DT with trajectory weighting (DT+W) outperforms DT by 8% on average, while RvS+W+C outperforms RvS+C by 5% on average. The performance gain is the most significant in medium-replay datasets (21% for DT on average, and 6% for RvS), where the data quality is generally lower. Moreover, trajectory weighting improves the success rate of RvS by 60% in the Antmaze tasks.
>
> > Would be interesting to see the significance of this phenomenon in actual benchmark datasets
>
> Figure 1 shows the effect of trajectory weighting in walker2d-medium-replay, which is an actual benchmark dataset in D4RL. In Appendix C.1, we additionally visualize the effect of trajectory weighting on the data distribution of walker2d-medium and walker2d-medium-expert.
>
> > Trajectory weighting leads to the problem of high variance
>
> We agree with the reviewer, and this is the bias-variance tradeoff we discussed in the paper. Learning from the original distribution leads to high bias, while completely ignoring the low-return trajectories leads to high variance. Our trajectory weighting upweights the trajectories by their returns, allowing one to balance this tradeoff. We also added a formal discussion on the bias-variance tradeoff in Appendix G.
>
> > The approximation of equation 5
>
> If I understand the question correctly, the reviewer is asking about the approximation of equation (6) to equation (5) (correct me if I’m wrong). As we mentioned in the paper, we cannot directly use (5) because the return is continuous and we only have a finite number of samples, so we have to discretize them into bins and reweight the bins instead. The number of bins is a hyperparameter in practice, and we found that using 20 bins works well across tasks.
>
> > Equation 5 in its current form is not completely novel
>
> MINs [1] was proposed for model-based optimization, while our work is proposed in the context of offline RL. In model-based optimization, the dataset consists of (x, y) pairs, whereas in offline RL we have a dataset of trajectories. Therefore, our trajectory weighting reweights the entire trajectories by their returns, as opposed to [1] which reweights individual data points. Moreover, [1] is based on GANs whereas we use vanilla supervised learning for our policy. We believe that adopting a known technique and showing it works in a new domain is an important contribution.
>
> > Why such a form of regularization is needed
>
> The need for conservatism arises naturally in return-conditioned methods – during evaluation, we need to choose a target return to condition on, which is a pain point as described in [2]. Specifically, we want to condition on a high return to achieve good performance, but if we condition on above a certain threshold, the performance degrades significantly. Our conservative regularization is proposed to prevent this performance crash problem, and has been shown to help achieve a much more stable performance (see Figure 4). We refer the reviewer to Section 4.2 for a detailed discussion.
>
> > Regularization schemes in offline RL has been well studied in recent past
>
> Previous works studied regularization in the context of value-based methods, in which they are mainly concerned with the extrapolation error in the value estimation. Therefore they either regularize the value function (e.g., CQL, IQL) or add an additional BC loss to prevent the policy from exploiting this error. In contrast, we study conservatism in the context of return-conditioned methods, in which we are concerned with the extrapolation error of the policy when conditioning on out-of-distribution returns. We therefore propose to perturb the returns during training to generate ood returns and regularize the predicted actions.
>
> [1] Aviral Kumar and Sergey Levine. Model inversion networks for model-based optimization. Advances in Neural Information Processing Systems, 33:5126–5137, 2020.
>
> [2] Emmons, S., Eysenbach, B., Kostrikov, I., & Levine, S. (2021). RvS: What is Essential for Offline RL via Supervised Learning?. arXiv preprint arXiv:2112.10751

---

> > ### Author Response · Authors · 2022-11-13
> > **Author response (Part 2 of 2)**
> >
> > > Results on Atari shows marginal improvements only
> >
> > Figure 9 shows that our method RvS+W+C (in red) is always above the original method (in blue). Table 6 compares the absolute performance, which shows that CWBC outperforms the original model by 72% on average on 4 tasks.
> >
> > > Additional experiments to make the paper more convincing
> >
> > In addition to the D4RL locomotion tasks, we benchmark CWBC on Atari with visual observations and Antmaze with sparse rewards. On Atari, CWBC outperforms the original model by 72% on 4 games on average (Table 6). On Antmaze, CWBC improves the success rate of the original model by 60% on 6 settings on average (Table 7). In total, we’ve evaluated CWBC on 3 benchmarks and 19 offline datasets, and CWBC has consistently improved the performance of the base method. We believe these experiments are sufficient in both quantity and diversity to show the effectiveness of our proposed framework.

---

> ### Author Response · Authors · 2022-11-16
> **Checking in**
>
> Thank you again for your review. We hope you have had a chance to read our response. We believe our reply addresses the weaknesses and questions raised by this review. Since the discussion period is short and drawing to a close soon, are there further questions or concerns we should discuss? Thank you.

---

> ### Author Response · Authors · 2022-12-03
> **Rebuttal reminder (Dec 3)**
>
> We thank the reviewer again for your review. It's been a few weeks since we posted our rebuttal, which we believe has addressed the weaknesses and questions raised by this review. We sincerely hope that the reviewer has had a chance to read the rebuttal and will respond to it soon. We are happy to address any remaining questions or concerns.

---

> ### Comment · Reviewer_vF8h · 2022-12-04
> **Thanks for the detailed response; Improving Score**
>
> Dear Authors,
>
> Thank you for the detailed response. I think your answers provided me with further clarification for the work, and I understand the significance of it better.
>
> I would like to clarify that following the responses, I understand some of my understanding of the paper may have been incorrect - and I think the work has more novelty that what I initially thought it to be.
>
> I would therefore like to improve the score of the paper.
>
> Apologies for the late response.

---

> > ### Author Response · Authors · 2022-12-06
> > **Author response**
> >
> > We thank the reviewer for appreciating the significance of the work and for improving the score.

---

### Official Review · Reviewer_VCrN · 2022-10-25

**Confidence:** 4
**Correctness:** 3
**Technical Novelty And Significance:** 2
**Empirical Novelty And Significance:** 2
**Recommendation:** 3

**Clarity, Quality, Novelty And Reproducibility:**

The main message of the paper is clearly conveyed. The novelty is a concern. The method is easy to reproduce and the code is provided.



**Strength And Weaknesses:**

Strengths:
1. The paper is easy to follow and well structured.
2. Experiment results support the authors claims.

Weaknesses:
1. Lack of novelty. The trajectory weighting technique is originated from Kumar & Levine, 2020. The solution for out-of-distribution (ood) returns is naïve and simple by adding positive random noise. The conditional BC offline RL algorithms are prior works, RvS and DT.
2. The contributions are not very strong in the current state of the paper. The paper develops two techniques specifically for DT and RvS, which are limited in contributions.
3. Improvement is not consistent via trajectory weighting and the final performance still underperforms SOTA methods.
4. Experiments are not sufficient. The authors validate their techniques on D4RL locomotion tasks, which are known to be easy tasks. Better add more experiments such as AntMaze, Adroit or Kitchen to make the claims more convincing.

Aviral Kumar and Sergey Levine. Model inversion networks for model-based optimization. Advances in Neural Information Processing Systems, 33:5126–5137, 2020.


**Summary Of The Paper:**

This paper proposes two techniques, trajectory weighting and conservative regularization, to overcome two challenges, the bias-variance tradeoff and out-of-distribution return condition, respectively. Empirical study is conducted on top of two conditional BC offline RL algorithms, Reinforcement Learning via Supervised Learning (RvS) and Decision Transformer (DT).

**Summary Of The Review:**

Overall, the depth of this contribution and novelty may not quite yet reach the high bar set for ICLR. The improvement is significant while the improved performance still underperforms the SOTA offline RL algorithms. Only D4RL locomotion tasks are experimented, and adding experiments on more challenging tasks would make the claims more convincing.

---

> ### Author Response · Authors · 2022-11-13
> **Author response**
>
> We thank the reviewer for the constructive feedback and for the recognition of the strengths of CWBC. We have updated the paper to include a formal discussion of the bias-variance tradeoff and more baseline results, which we marked in red. We answer each of the reviewer's concerns below.
>
> > The trajectory weighting technique is originated from Kumar & Levine, 2020
>
> MINs [1] was proposed for model-based optimization, while our work is proposed in the context of offline RL. In model-based optimization, the dataset consists of (x, y) pairs, whereas in offline RL we have a dataset of trajectories. Therefore, our trajectory weighting reweights the entire trajectories by their returns, as opposed to [1] which reweights individual data points. Moreover, [1] is based on GANs whereas we use vanilla supervised learning for our policy. We believe that adopting a known technique and showing it works in a new domain is an important contribution.
>
> > The solution for out-of-distribution (ood) returns is naïve and simple
>
> The simplicity of our conservative regularization is a strength and not a weakness, as it can be generally and easily applied to any existing return-conditioned method. Moreover, as shown in our experiments across different benchmarks, conservative regularization has consistently improved and stabilized the performance of the original model.
>
> > The conditional BC offline RL algorithms are prior works
>
> Our work aims to point out the major challenges faced by return-conditioned methods and propose simple solutions for these challenges. That’s why we apply our framework to existing methods and show that it improves the performance of these methods.
>
> > The paper develops two techniques specifically for DT and RvS, which are limited in contributions.
>
> We respectfully disagree. The techniques are proposed in the general context of return-conditioned methods, and can be applied to learn any existing return-conditioned policy. We choose DT and RvS as they are two of the most commonly used algorithms.
>
> > Improvement is not consistent via trajectory weighting
>
> For DT, trajectory weighting improves the performance in 6/9 mujoco tasks, while performing very similarly in others. For RvS, the performance gain is not consistent because of the performance crash problem, for which we have provided a detailed discussion in section 4.2. When used in conjunction with the conservative regularization, the improvement gain via trajectory weighting is consistent across mujoco, atari, and antmaze benchmarks.
>
> > The final performance still underperforms SOTA methods
>
> CWBC is SOTA among BC-based methods in Mujoco, and is the best overall method in Antmaze. In tasks where we do not achieve SOTA, CWBC has significantly narrowed the gap between existing conditional BC methods and the SOTA value-based methods. Moreover, we believe that understanding and proposing solutions for the shortcomings of current methods is a good contribution to the community.
>
> > Experiments are not sufficient
>
> In addition to D4RL locomotion tasks, we already benchmarked our proposed method on Atari games (with visual observations) and Antmaze (with sparse rewards) in Appendix D and Appendix E. CWBC outperforms the original model by 72% and 60% on Atari and Antmaze, respectively. In total, we’ve evaluated CWBC on 3 benchmarks and 19 offline datasets, and CWBC has consistently improved the performance of the base method. We believe these experiments are sufficient in both quantity and diversity to show the effectiveness of our proposed framework.
>
> [1] Aviral Kumar and Sergey Levine. Model inversion networks for model-based optimization. Advances in Neural Information Processing Systems, 33:5126–5137, 2020.

---

> > ### Comment · Reviewer_VCrN · 2022-12-01
> > **Re: Response**
> >
> > Thank for the response. I still have some concerns unsolved.
> >
> > This paper proposes two techniques for return-conditioned BC-based offline RL algorithms: the trajectory reweighting and the conservative regularization via adding return noises. These two are simple and naïve, which could be considered as training tricks and are specifically designed for return-conditioned BC-based algorithms. This is why I think the contributions are not very strong in the current state of the paper.
> >
> > Moreover, the empirical performance is not very promising. Even with the two techniques, the RvS+W+C and DT+W still underperform the SOTA CQL and IQL on D4RL locomotion benchmark. Note that CQL and IQL have much smaller number of parameters in their policy network. Here I apologize that I missed your AntMaze results in Table 7 in Appendix E during my first review. However, after reading through it, I suspect the authors cherry-picked the baseline methods shown in Table 7. I am not sure why the IQL results were not cited here for comparison while IQL results are considered in Table 2. (IQL as a BC baseline should significantly outperform all the others in Table 7 with an average score around 50).
> >
> > I'm therefore keeping my score as-is.

---

> > > ### Author Response · Authors · 2022-12-02
> > > **Author response**
> > >
> > > Thank you for your response. We answer each of the reviewer's remaining concerns below.
> > > > These two are simple and naive which can be considered as training tricks
> > >
> > >  We strongly believe that the simplicity of the paper is a strength and not a weakness, as it can be applied to any existing return-conditioned methods. It is not naive because it has consistently shown effectiveness in all scenarios we considered.
> > >
> > > > Specially designed for return-conditioned BC-based algorithms
> > >
> > > This is one of the major classes of algorithms for offline RL, and we have shown that our proposals significantly improve their performance, which we believe is a strong contribution to the field.
> > >
> > > > RvS+W+C and DT+W still underperform the SOTA CQL
> > >
> > > We believe not all works must achieve SOTA to make an impact. Rejecting a paper for not achieving SOTA is also not a valid reason as per the ICLR reviewing guidelines: https://iclr.cc/Conferences/2023/ReviewerGuide . The aim of our paper is to study the challenges faced by return-conditioned methods, which is a major class of offline RL algorithms, and propose simple yet effective solutions to address them. Our methods improve on these methods consistently. Even DT and RvS did not beat SOTA methods such as CQL and IQL when first proposed, yet were accepted by venues such as NeurIPS and ICLR respectively.
> > >
> > > > Note that CQL and IQL have much smaller number of parameters in their policy network.
> > >
> > > We do not understand why the number of parameters is important here, because the architectures and hyperparameters of these methods have been tuned to achieve the best performance. Moreover, RvS also only uses an MLP policy, so there is no significant difference in parameter count.
> > >
> > > > I am not sure why the IQL results were not cited here for comparison
> > >
> > > We got the baseline numbers from the RvS paper which did not include IQL. We’ll include IQL numbers in the updated version. Again, no where in the paper did we state our goals as achieving SOTA and in fact, we have acknowledged that currently, on average, *all* return base methods lag behind value-based methods currently. We believe the fact that our paper shrinks this gap between value and return based methods is a sign of progress and significant for the community.

---

> ### Author Response · Authors · 2022-11-16
> **Checking in**
>
> Thank you again for your review. We hope you have had a chance to read our response. We believe our reply addresses the weaknesses and questions raised by this review. Since the discussion period is short and drawing to a close soon, are there further questions or concerns we should discuss? Thank you.

---

### Official Review · Reviewer_rbiQ · 2022-10-26

**Confidence:** 4
**Correctness:** 3
**Technical Novelty And Significance:** 2
**Empirical Novelty And Significance:** 2
**Recommendation:** 6

**Clarity, Quality, Novelty And Reproducibility:**

The paper is very well written and easy to follow.

The proposed method is novel in the context of return-conditioned method, but I would have like to see a comparison to the simple baseline commonly used in value-based methods (mentioned previously).

The pseudo-code is included so I believe the algorithm is reproducible.

**Strength And Weaknesses:**

Strengths

1. The paper is relatively well-written and easy to follow.

2. The proposed idea is intuitive, whose benefits are demonstrated on two different classes of return-conditioned methods (DT and RvS).

Weaknesses

1. The paper makes repeated reference to a "bias-variance trade-off" to motivate their methods. AFAIK, what this trade-off means exactly, and a theoretical or rigorous analyses as to why their method leads to a more favorable trade-off are missing completely.

2. Skewed distribution of returns is a well-known problem in RL, and common technique to tackle this problem is to ensure each batch consists of equal amount of high return and low return episodes (so equivalently, use two bins only). I would have liked to see a comparison to such a simple baseline.

3. My biggest concern is the motivation behind studying return-condition method? Because AFAIK, they under-performs value-based methods, especially on tasks that require stitching (such as antmaze navigation and notshown in the paper). Even in table 2, the performance of return-conditioned methods are under-whelming. I am not saying that the paper should not be accepted because it does not out-perform state-of-the-art, but there should be a reason as to why the class of return-conditioned method is interesting relative to value-based methods.


**Summary Of The Paper:**

The paper identifies the skewed distribution of returns in the offline dataset as a challenge to return-conditioned methods, and proposes to weight the trajectories in the dataset based on their returns.


**Summary Of The Review:**

I recommend a weak reject, because it is unclear why we should be interested in return-conditioned methods, especially when they face their own sets of unique challenges, and when enhanced with proposals such as the ones in this paper, still under-perform value-based methods. What are their unique selling points?

---

> ### Author Response · Authors · 2022-11-13
> **Author response (Part 1 of 2)**
>
> We thank the reviewer for the constructive feedback and for the recognition of the strengths of CWBC. We have updated the paper to include a formal discussion of the bias-variance tradeoff and more baseline results, which we marked in red. We answer each of the reviewer's concerns below.
> > A theoretical or rigorous analyses as to why their method leads to a more favorable trade-off are missing completely
>
> We have added a formal discussion and analysis of the bias-variance tradeoff in Appendix G.
>
> > I would have liked to see a comparison to such a simple baseline.
>
> The reviewer’s suggested technique is a special case of our trajectory weighting technique, where we have 2 bins and they are equally weighted. The table below shows the performance of DT when trained on the original distribution, with the technique suggested by the reviewer (2 equally weighted bins), and with trajectory weighting. Trajectory weighting achieves the best performance in all 3 tasks, and also has the lowest variance.
>
> |                           | Original distribution | 2 equally weighted bins | Trajectory weighting |
> |---------------------------|-----------------------|-------------------------|----------------------|
> | walker2d-medium-replay    | 53.4 +- 12.2          | 59.05 +- 10.74          | 60.5 +- 8.9          |
> | hopper-medium-replay      | 56.4 +- 20.1          | 68.1 +- 12.55           | 76.9 +- 5.9          |
> | halfcheetah-medium-replay | 34.5 +- 4.2           | 31.09 +- 2.8            | 36.9 +- 0.2          |
>
> Intuitively, if we only use two bins, and the return range is wide enough, the high-return bin still contains a lot of low-quality trajectories. Using more bins allows us to have a more fined categorization of the quality of the trajectories. Moreover, this technique treats the two bins equally, which can actually downweight the high-return bin if it has more weights in the original distribution. This is the case in halfcheetah-medium-replay, and that’s why this technique underperforms the original distribution. In contrast, our trajectory weighting always upweights the high-return bins exponentially.
>
> > Results on stitching-required environments and comparison with value-based methods
>
> We already included experiments for Antmaze tasks in Table 7, appendix E, and have also updated the table to compare with the value-based methods. CWBC actually outperforms CQL Antmaze, and achieves the best performance among all considered methods in this task.
>
> > Even in table 2, the performance of return-conditioned methods are under-whelming
>
> Table 2 shows that outperforms TD3+BC, on par with IQL, and only slightly underperforms CQL.

---

> > ### Author Response · Authors · 2022-11-13
> > **Author response (Part 2 of 2)**
> >
> > > Motivation behind studying return-condition methods
> >
> > Return-conditioned methods have great potential and benefits in solving offline RL:
> > - Compared to value-based methods, they are much simpler and more stable to train.
> > - This paradigm introduces a new view that connects offline RL with supervised learning and generative modeling, allowing us to utilize many exciting advances in these areas, such as diffusion models for more accurate long-horizon planning [2].
> > - A major benefit of return-conditioned methods is the unprecedented scalability with respect to both data and model size. For example, Multi-game DT [4] was the first to achieve super-human performance in all Atari games using a single neural network, and its performance increases with respect to model scaling, whereas the other methods either saturate or have much slower performance growth.
> > - Many recent publications at ML venues have been proposed to better understand return-conditioned RL by drawing connections to hindsight information matching [5], studying their exploration strategies in the online setting [6], and deriving theoretical guarantees [7, 8]. Our work contributes to understanding their practical limitations, and proposes simple solutions to address them.
> >
> > [1] Emmons, S., Eysenbach, B., Kostrikov, I., & Levine, S. (2021, September). RvS: What is Essential for Offline RL via Supervised Learning?. In International Conference on Learning Representations.
> >
> > [2]  Janner, M., Du, Y., Tenenbaum, J. &amp; Levine, S.. (2022). Planning with Diffusion for Flexible Behavior Synthesis. Proceedings of the 39th International Conference on Machine Learning, in Proceedings of Machine Learning Research 162:9902-9915 Available from https://proceedings.mlr.press/v162/janner22a.html.
> >
> > [3] Janner, M., Li, Q., & Levine, S. (2021). Offline reinforcement learning as one big sequence modeling problem. Advances in neural information processing systems, 34, 1273-1286.
> >
> > [4] Lee, K. H., Nachum, O., Yang, M., Lee, L., Freeman, D., Xu, W., ... & Mordatch, I. (2022). Multi-Game Decision Transformers. Advances in neural information processing systems, 35.
> >
> > [5] Furuta, H., Matsuo, Y., & Gu, S. S. (2021, September). Generalized Decision Transformer for Offline Hindsight Information Matching. In International Conference on Learning Representations.
> >
> > [6] Zheng, Q., Zhang, A. & Grover, A.. (2022). Online Decision Transformer. Proceedings of the 39th International Conference on Machine Learning, in Proceedings of Machine Learning Research 162:27042-27059 Available from https://proceedings.mlr.press/v162/zheng22c.html.
> >
> > [7] Kumar, A., Hong, J., Singh, A., & Levine, S. (2021, September). Should I Run Offline Reinforcement Learning or Behavioral Cloning?. In International Conference on Learning Representations.
> >
> > [8] Brandfonbrener, D., Bietti, A., Buckman, J., Laroche, R., & Bruna, J. (2022). When does return-conditioned supervised learning work for offline reinforcement learning?. Advances in neural information processing systems, 35.

---

> > ### Comment · Reviewer_rbiQ · 2022-12-02
> > **Results on AntMaze**
> >
> > Hi authors,
> >
> > My apologies for the late reply.
> >
> > I will add a longer form reply soon, but one thing that jumps out to me when I saw your replies is that I am not sure I am convinced by the results in Table 7 in the paper.
> >
> > The reason is because: In Table 7, CQL is shown to have zero performance across many AntMaze tasks. But Table 1 in (Emmons et al., 2021) actually shows two sets of results for CQL, and CQL-p has much higher performance that what is shown in Table 7 in your paper. So I am a bit concerned about what you are actually claiming in Table 7 ...

---

> > > ### Author Response · Authors · 2022-12-02
> > > **Author response: Results on Antmaze**
> > >
> > > Thank you for your response.
> > >
> > > Table 1 in [1] has two sets of CQL results: CQL-p and CQL-r, in which CQL-p denotes the performance published in the original paper [2], while CQL-r denotes the performance replicated by the RvS authors by using the CQL open-source code. We chose to report CQL-r because according to [1], CQL-p was obtained using the v0 version of the datasets, while CQL-r, RvS, and we used the v2 version. We will include both CQL-p and CQL-r to avoid confusion in the updated version of the paper.
> > >
> > > [1] Emmons, Scott, et al. "RvS: What is Essential for Offline RL via Supervised Learning?." International Conference on Learning Representations. 2021.
> > > [2] Kumar, Aviral, et al. "Conservative q-learning for offline reinforcement learning." Advances in Neural Information Processing Systems 33 (2020): 1179-1191.

---

> > > > ### Comment · Reviewer_rbiQ · 2022-12-03
> > > > **Increased my score to 6**
> > > >
> > > > I have increased my score to 6, since the idea proposed is simple, novel (afaik, even if previous works were used to motivate the technique, that is okay) and demonstrates performance benefit for the methods considered in the context of the paper. The authors have addressed some of the concerns that I raised previously.
> > > >
> > > > I am still uncertain about the results though because:
> > > >
> > > > 1. Re-weighting the distribution can also be done for value-based methods, so it is unclear if value-based methods would also perform well when re-weighted.
> > > >
> > > > 2. Regarding v0 and v2 differences, the gap in performance of CQL indicates to me a bug in obtaining CQL performance. So I am not convinced by the results on AntMaze.
> > > >
> > > > 3. A major benefit of return-conditioned methods is the unprecedented scalability with respect to both data and model size. For example, Multi-game DT [4] was the first to achieve super-human performance in all Atari games using a single neural network, and its performance increases with respect to model scaling, " whereas the other methods either saturate or have much slower performance growth. " => I don't think there is enough evidence for the statement between the quotation marks to be true.

---

> > > > > ### Author Response · Authors · 2022-12-03
> > > > > **Author response**
> > > > >
> > > > > We thank the reviewer for appreciating the simplicity and effectiveness of our proposed method and for raising the score. We address the remaining concerns below.
> > > > > > Regarding v0 and v2 differences, the gap in performance of CQL indicates a bug in obtaining CQL performance
> > > > >
> > > > > We will contact RvS and CQL authors to clarify the main reason leading to different performances with respect to v0 and v2. We will also add CQL-p to Table 7 for completeness.
> > > > >
> > > > > >  Re-weighting the distribution can also be done for value-based methods
> > > > >
> > > > > We agree with the reviewer and will investigate reweighting for value-based methods in future work. We will explicitly mention this in the updated version of the paper.

---

> ### Author Response · Authors · 2022-11-16
> **Checking in**
>
> Thank you again for your review. We hope you have had a chance to read our response. We believe our reply addresses the weaknesses and questions raised by this review. Since the discussion period is short and drawing to a close soon, are there further questions or concerns we should discuss? Thank you.

---

### Author Response · Authors · 2022-11-17
**Rebuttal reminder**

We thank the reviewers again for the reviews. It has been a few days since we posted our rebuttal, and we sincerely hope you have had a chance to read our response. We believe our reply has addressed the concerns and answered the questions raised by the reviewers. As the discussion period is short and about to end soon, are there any further questions or concerns we should discuss?

---

### Author Response · Authors · 2022-11-18
**Rebuttal reminder (2)**

We sincerely hope the reviewers have had a chance to read our rebuttal, which we believe has addressed the major concerns and questions raised in the reviews. As the discussion period is about to end soon, are there any further concerns or questions that we can discuss?

---

### Author Response · Authors · 2022-11-30
**Rebuttal reminder (Nov 30)**

Dear reviewers and AC,

It's been a few weeks since we posted our rebuttal, and we sincerely hope that the reviewers have had a chance to read our rebuttal and will respond to it soon. We are happy to answer any questions or concerns not yet addressed in the rebuttal.

---

### Decision · Program_Chairs · 2023-01-20

**Decision:**

Reject

**Justification For Why Not Higher Score:**

The paper shows promise but does not make a large enough contribution to warrant acceptance.

**Justification For Why Not Lower Score:**

N/A

**Metareview: Summary, Strengths And Weaknesses:**

The paper addresses the data sparseness problem in offline reinforcement learning. Two components are proposed: trajectory reweighting and conservatism introduced through a proposed regularizer. These are evaluated individually and in combination on MuJoCo and Atari tasks.

Initial reviews for this paper appreciated the intuitively appealing idea presented here, along with the empirical benefits the authors demonstrate.

Several concerns were raised in the initial reviews and the subsequent discussion. Reviewers noted the need to clearly define concepts such as the bias-variance tradeoff referred to in the paper. They asked for a clear motivation for the focus on return-conditioned approaches, and additional empirical validation. Many of the reviewer concerns were addressed by the authors during the rebuttal: clarity was improved on several points, motivation was discussed, and the authors added an additional empirical comparison with a simple weighting scheme.

A key remaining concern, and subject of much discussion, is the novelty and significance of the contribution of this work. Some reviewers have argued that, given that the two proposed modifications are relatively simple and widely known in the research community, both novelty and the size of the contribution are limited. The authors argue that simple approaches can be the basis of very valuable contributions, as can the study of known approaches in new contexts or domains.

After reviewing the paper and discussion in detail, the AC agrees with the perspective that the size of the contribution of the present paper is limited. It is true that simple and/or known approaches can be the basis of valuable contributions. However, to achieve this, the novel insights must have an appropriate depth and provide a corresponding value to the community. Here, depth of analysis does not go very far beyond comparisons of the obtained scores. It would be interesting to see additional analysis to empirically demonstrate the mechanisms behind the obtained performance, i.e., the how and why of the role that the proposed modifications play. Just to give one example, a key contribution is the introduction of conservatism via a regularizer that injects random noise on the return. In RvS paper (one of two baseline approaches used in this work), a whole section of the paper was dedicated to "ARCHITECTURE, CAPACITY, AND REGULARIZATION", identifying this interplay as essential for the performance of these methods. The present work proposes an alternative regularizer, and show empirically that it performs well on the tasks studied. However, in the present paper we gain little further insight on questions such as how the behavior of the proposed regularizer differs or aligns with that observed in the original work.

In sum, the paper presents an appealing approach but does not yet demonstrate a contribution that warrants an accept recommendation.